# Assessing economic impacts of future GLOFs in Nepal's Everest region under different SSP scenarios using three-dimensional simulations

Wilhelm Furian[1] and Tobias Sauter[1]

[1]Climatology Lab, Geography Department, Humboldt-Universität zu Berlin, Berlin, Germany

*Correspondence to*: Wilhelm Furian (wilhelm.furian.1@geo.hu-berlin.de)

**Abstract**

This study investigates simulated glacial lake outburst floods (GLOFs) at five glacial lakes in the Everest region of Nepal using the three-dimensional model OpenFOAM. It presents the evolution of GLOF characteristics in the 21st century considering different moraine breach scenarios and two Shared Socioeconomic Pathways scenarios. The results demonstrate that in low-magnitude scenarios, the five lakes generate GLOFs that inundate between 0.35 and 2.23 $km^2$ of agricultural land with an average water depth of 0.9 to 3.58 meters. These GLOFs reach distances of 59 to 84 km, affect 30 to 88 km of roads or trails, and inundate 183 to 1,699 buildings with 1.2 to 4.9 meters of water. In higher scenarios, GLOFs can extend over 100 km and also affect larger settlements in the foothills. Between 80 and 100 km of roads, 735 to 1,989 houses and 0.85 to 3.52 $km^2$ of agricultural land could be inundated, with average water depths of up to 10 meters. The high precision of the 3D flood modeling, with detailed simulations of turbulence and viscosity, provides valuable insights into 21st-century GLOF evolution, supporting more accurate risk assessments and effective adaptation strategies.

**Short summary**

Glacial lake outburst floods (GLOFs) continue to threaten high-mountain communities in Nepal. We simulate potential GLOF events from five glacial lakes in the Everest region during the 21st century using a 3D flood model and several breach and SSP scenarios. Large GLOFs could extend over 100 km and inundate 80 to 100 km of roads and trails, 735 to 1,989 houses and between 0.85 and 3.52 $km^2$ of agricultural land. The results help to assess the changing GLOF impacts and support more accurate risk assessments.

## 1. Introduction

Climate change induces elevated temperatures and accelerated glacier melt across extensive areas of High Mountain Asia (Hock et al., 2019; Shean et al., 2020). One of the most apparent consequences of this glacier melt is the formation of new glacial lakes or the expansion of existing lakes. Driven by higher glacier runoff or increasing precipitation, the water can accumulate behind moraines or in depressions exposed by the retreating glaciers (Haritashya et al., 2018; Farinotti et al., 2019b). Glacial lakes present opportunities for tourism, hydropower production, and water supply (Farinotti et al., 2019b; Haeberli et al., 2016). However, they also significantly increase the hazard and risk potential for downstream populations and infrastructure.

The abrupt release of water due to a moraine failure has the potential to evolve into a catastrophic debris flow, which can extend over long distances. These glacial lake outburst floods (GLOFs) have been recorded to exceed 100 kilometers (Allen et al., 2016) and, in extreme cases, even 500 kilometers (Hewitt and Liu, 2010). Consequently, these GLOFs represent one of the most hazardous natural phenomena in high mountain regions, posing a significant threat to both human populations and infrastructure (Ahmed et al., 2021). In High Mountain Asia (HMA), these floods have resulted in the highest fatality rates globally (Carrivick and Tweed, 2016). It is therefore imperative to compile data on the potential extent of GLOF events at specific lakes. To this end, it is essential to examine a comprehensive range of scenarios, including a variety of potential dam breach cross-sections and different climate and population development scenarios, not only for the current lakes but also at multiple points throughout the 21st century (Haeberli et al., 2016; Veh et al., 2020). This information can assist in the identification of particularly vulnerable infrastructure and the detection of potential hotspots where adaptation measures are especially required.

The study of dam-break problems and subsequent GLOF events is extensive and ranges from global studies on GLOF potential (Zheng et al., 2021a; Taylor et al., 2023) to simulated replicas of individual historical GLOFs (Anacona et al., 2018; Washakh et al., 2019). There are also several hydrodynamic simulations of potential GLOFs at particularly dangerous lakes (Somos-Valenzuela et al., 2016; Pandey et al., 2022). However, in order to support the sustainable development of high-mountain communities, it is also necessary to consider future scenarios and investigate the potential changes in GLOFs at specific locations over the course of the 21st century. To date, this has been accomplished in a limited number of studies that have modeled worst-case scenarios of GLOFs in addition to the status quo simulations (Sattar et al., 2021; Majeed et al., 2021; Allen et al., 2022). To the best of our knowledge, only one study has thus far investigated the influence of different climate scenarios on future GLOF characteristics (Zheng et al., 2021a). However, due to the large scope of their study, they employed a highly simplified routing algorithm for the GLOF flow paths.

The popularity of numerical modeling in GLOF research has increased in recent years due to its flexibility, cost-effectiveness, and the ability to process complex data with ever-increasing computing capabilities. Furthermore, the

accessibility of a multitude of computational fluid dynamics (CFD) software with user-friendly interfaces and effective processing tools has enabled the extensive application of numerical models in flood hazard analysis. GLOF events have been predominantly investigated using one-dimensional (1D) or, more recently, two-dimensional (2D) hydrodynamic models. 2D approaches are demonstrated by Anacona et al. (2015) and Sattar et al. (2021), who used the HEC-RAS model. Lala et al. (2018) employed the BASEMENT model, while Somos-Valenzuela et al. (2015) and Wang et al. (2022b) utilized the FLO-2D model to investigate the range and inundation depths of GLOF events.

Depth-averaged 2D models solve the shallow water equations (SWE) using various numerical methods to predict hydraulic quantities like maximum flood depth and velocity, assuming negligible vertical acceleration and minimal surface curvature. However, in steep, mountainous terrains, the hydrostatic pressure assumption is violated, especially during initial dam-break stages or when flood waves navigate bends and obstacles (Maranzoni and Tomirotti, 2023). In contrast, 3D hydrodynamic modeling offers several improvements over 2D models for flooding simulations, effectively addressing the inherent limitations of 2D models especially in strongly perturbed flow (Dellinger et al., 2024; Lee et al., 2024). Unlike the latter, 3D models calculate pressure fields, include vertical fluid acceleration, and describe vertical velocity variations, capturing the effects of flow curvature. They also simulate vertical turbulence and spiral flows, ensuring precise modeling of flow impacts against obstacles (Larocque et al., 2013). This results in more accurate predictions of dam-break dynamics, particularly around structures and in complex terrains, improving flood hazard assessments (Akgun et al., 2023). Despite their superiority in modeling dam failure and inundation characteristics in steep terrain, the use of 3D models involves higher computational costs and requires more complicated initial setup and parameter implementation. This may be the reason why only a few studies have used 3D models to investigate debris flows following a dam-break in mountainous areas, such as Zhuang et al. (2020) or Gao et al. (2024). To our knowledge, no research has been published so far detailing the use of a full 3D CFD model to simulate a GLOF event.

In this study, we model hypothetical GLOF events at five glacial lakes in the Everest region of Nepal using a fully three-dimensional CFD model. In comparison to entire HMA, Nepal has experienced a more pronounced warming trend, and the reduction in ice mass, the loss of permafrost and the upward shift of the equilibrium line of altitude (approximately 20 meters per decade) contribute to Nepal's heightened vulnerability to GLOF events (Kraaijenbrink et al., 2017; Khadka et al., 2023; Baral et al., 2023). We investigate how GLOFs could change over the course of the 21$^{st}$ century in various outburst scenarios for each time step. To consider future socio-economic and political scenarios, we utilize two pathways from the Shared Socioeconomic Pathway (SSP) scenarios (Meinshausen et al., 2020). For our CFD model, we have chosen the open-source model OpenFOAM (Weller et al., 1998). Our results can support more precise assessments of future GLOFs in the region, aiding the implementation of critical monitoring and investigative measures at potentially hazardous lakes, the development of early warning systems where applicable, and the adoption of additional GLOF adaptation strategies.

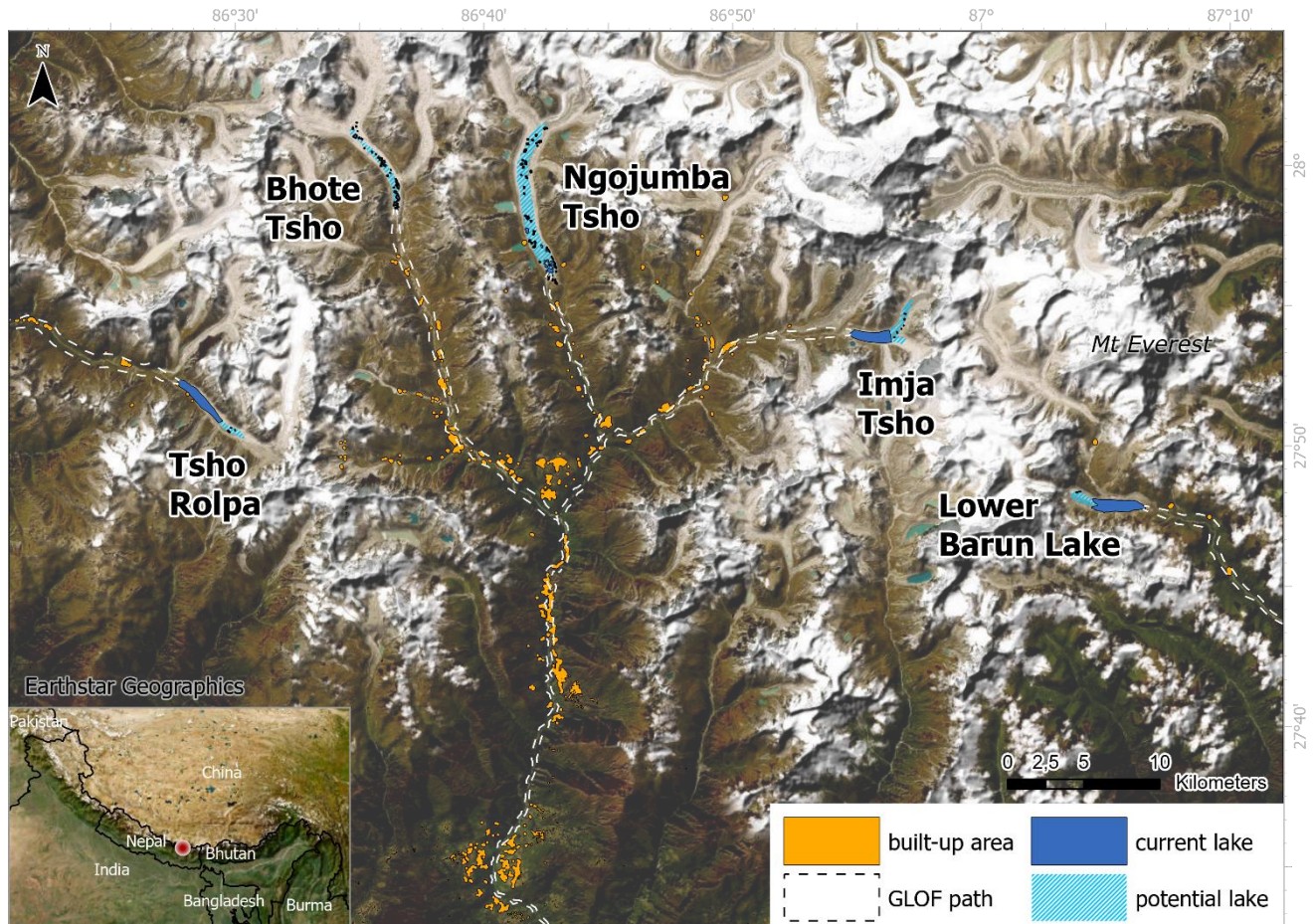

**Figure 1: Overview of the study area, the Everest region in Nepal. Tsho Rolpa, Imja Tsho and Lower Barun Lake are depicted in their current extent in darker blue, while numerous supraglacial lakes are present at the locations of both potential future lakes (Ngojumba and Bhote Tsho). Light blue dashed areas indicate the maximum lake extent in 2100, following Furian et al. (2022). Orange areas show the built-up areas in the vicinity (© OpenStreetMap, OSM contributors, 2024), buffered for visibility. Dashed white lines indicate the flow path of a GLOF event.**

## 2. Study area

The Everest region in northeastern Nepal is an important part of the country's tourism industry, contributing significantly to the local and national economy through tourism revenues and employment. In addition to its economic impact, the region is culturally significant and includes parts of the Sagarmatha National Park, a UNESCO World Heritage Site (Sun and Watanabe, 2021). Due to its great importance for tourism, infrastructure development is expected to continue to encroach into the higher elevations of the region (Neupane et al., 2024). In the event of a GLOF, this would mean that an increasing number of people and infrastructure would be affected (Zhang et al., 2024). This study focuses on simulated GLOFs from five lakes in the region, all located in the center of the Koshi Basin—a sub-basin of the Ganges River—where glacial lakes are expanding at the fastest

rate in the Nepalese Himalaya (Khadka et al., 2018). Therefore, glacial lakes and associated GLOF risks have received considerable research attention in this area (e.g., Byers et al., 2019; Bajracharya et al., 2020; Khadka et al., 2021; Gouli et al., 2023). Figure 1 shows the lakes selected for this study in their current and potential future states, possible GLOF paths, and surrounding developed areas.

Imja Tsho, one of Nepal's fastest growing lakes, has exhibited a significant increase in surface area, from 0.81 km² in 1997 to 1.56 km² in 2020 (Gupta et al., 2023). It is one of the most studied lakes in Nepal, and due to its high damage potential in case of a GLOF, the water level was lowered in 2016. At that time, it was also determined that if the lake continues to expand eastward at its current pace, in 2035 it will reach an area where substantial avalanches could enter the lake from the surrounding mountains (Rounce et al., 2017). Lala et al. (2018) propose that the likelihood of a catastrophic GLOF at Imja Tsho has diminished as a consequence of the lowered water level. Nevertheless, the lake remains one of the most dangerous in the region, with an estimated 100,000 individuals potentially impacted in the event of a GLOF (Bajracharya et al., 2020). While the rate of expansion has slowed, the presence of ice beneath the approximately 55-meter-high end moraine and the merging of supraglacial ponds continue to represent significant concerns (Bajracharya et al., 2020; Chen et al., 2025).

With an area of 1.53 km², the Tsho Rolpa is also regarded as one of the most dangerous lakes in Nepal (Chen et al., 2022) and has been the subject of intense study (e.g., Sakai et al., 2000; Sherry and Curtis, 2017; Kayastha and Maskey, 2024). In the event of a GLOF, approximately 150,000 individuals could be impacted (Bajracharya et al., 2020). Consequently, the water level was reduced in 2000 (Shrestha et al., 2007). However, the lake is expanding rapidly due to the retreat of the calving source glacier, with a steep and thin moraine, and a hanging lake in the tributary glacier adding to the danger. As a consequence of its accelerated growth, glacier modeling indicates that the lake will reach its maximum area (approximately 2.5 km²) within the next decade (Furian et al., 2022).

Lower Barun Lake, also one of the most dangerous lakes in Nepal, formed in the 1960s from several supraglacial ponds. Subsequently, the lake has expanded continuously due to the retreat of the glacier, and it currently covers an area of approximately 2 km² (Haritashya et al., 2018). The lake presents a significant danger due to the ongoing expansion of the glacier, which also serves as a calving source, and the considerable surrounding mountain relief, which poses a substantial risk of landslides and ice avalanches (Bajracharya et al., 2020; Maskey et al., 2020; Watson et al., 2020). The end moraine at Lower Barun is undergoing thermokarstic evolution, which could either gradually lower the lake due to ice melt or transform the moraine into a precarious, narrow structure similar to the moraine at Tsho Rolpa, thereby increasing the risk of moraine failure (Sattar et al., 2021).

The two potential proglacial lakes at Ngojumba Glacier and Bhote Kosi Glacier have the potential to form within the next decade, according to glacier modeling (Furian et al., 2022). While the actual lake formation is uncertain and depends on a multitude of factors (e.g., glacial retreat, moraine stability, debris cover), an abundance of supraglacial ponds and lakes at

both locations serves to reinforce this possibility. The lake at Bhote Kosi could potentially reach a size of 2.6 km² by the year 2100, should a stable moraine develop. At Ngojumba, the estimated subglacial morphology would allow for the formation of

a substantial glacial lake with an area exceeding 8 km² and a volume of more than $700 \times 10^6$ m³. As the future composition of the moraines at both sites remains undetermined, we utilize the calculated lake areas as a potential maximum in our simulations and any GLOFs as a worst-case scenario for these lakes. In this study, we refer to these future lakes as Ngojumba Tsho and Bhote Tsho.

Some studies have been conducted using 1D or 2D modeling to investigate the results of a GLOF at the Lower Barun

Lake (Sattar et al., 2021; Mandal et al., 2025), the Imja Tsho (Somos-Valenzuela et al., 2015; Lala et al., 2018), and the Tsho Rolpa (Chen et al., 2022; Kayastha and Maskey, 2024). In the event of a GLOF, these studies indicate that several settlements downstream could be affected in each case, as well as large areas of farmland, several bridges and hydropower plants (HPP).

## 3. Data

In this study, we employ the CFD model OpenFOAM to simulate hypothetical GLOFs and examine their potential evolution

in the 21$^{st}$ century under a range of outburst scenarios and socio-economic pathways. To achieve this, we rely on a variety of input data. For the digital elevation model (DEM), we chose the radiometric terrain corrected ALOS PALSAR dataset with a resolution of 12.5 meters (ASF DAAC, 2015). The glaciers extent and their future terminus locations are derived from a previously published dataset (Furian et al., 2022). Furthermore, a high-resolution bathymetric dataset provided by the UK Centre for Ecology & Hydrology was employed to enhance the resolution of Tsho Rolpa and its surrounding area (Maharjan

et al., 2021). In the case of Imja Tsho and the Lower Barun Lake, bathymetries were adapted from those originally presented by Haritashya et al. (2018) and Somos-Valenzuela et al. (2013).

As GLOF triggers are not directly simulated in this study, we utilize previously published data (Furian et al., 2021) as a first-order evaluation of the hazard of GLOFs caused by mass-movement impacts into the lake. They provide lake hazard levels by estimating the likelihood of detached material from surrounding slopes impacting the lake. The mechanical and

thermal regime of these slopes is significantly influenced by the state of their frozen water content (GAPHAZ, 2017). Accordingly, we use data on the current permafrost status in the region (Obu et al., 2018). Also, we utilize ISIMIP2b output data on future permafrost (Burke et al., 2023). In summary, the model outputs suggest a gradual decline of permafrost until midcentury under SSP2, followed by a potential for gradual recovery. In contrast, for SSP5, both soil temperature and thaw depth are expected to increase until, by approximately 2070, the permafrost in this region is expected to have disappeared, with the

potential exception of some isolated remnants. By integrating both permafrost data sets with the slope hazard, a lake hazard score is assigned to each lake. This score reflects the changing predisposition of each lake for mass-movement impacts capable of triggering a GLOF throughout the 21$^{st}$ century.

To consider future socio-economic developments, two SSP scenarios were selected for analysis: the SSP2 and the SSP5 (Meinshausen et al., 2020). The "Middle of the Road" scenario, designated as SSP2, portrays a world with an additional radiative forcing of 4.5 W/m² in 2100. In contrast, SSP5, the "Taking the Highway" scenario, posits a scenario with an additional radiative forcing of 8.5 W/m² by the year 2100. Although the SSP5 scenario is considered the most representative of global development during the past two decades (Schwalm et al., 2020), the SSP2 scenario is perceived by numerous researchers to be the more probable pathway scenario (Hausfather and Peters, 2020). Accordingly, in the present study, the SSP2 is employed as a conservative lower estimate, while the SSP5 represents the "worst case" scenario.

To assess the potential GLOF risks to human life and infrastructure, we combine the World Settlement Footprint (WSF) dataset (Marconcini et al., 2021) and OpenStreetMap (OSM) data (OSM contributors, 2024). These combined data provide detailed insights into the location of residential and other human-made structures, as well as the land cover and land use in the vicinity of GLOF pathways. In contrast to previous studies, we also incorporate population forecast datasets in our analysis. These include ISIMIP2b input data (Piontek and Geiger, 2017), data from the World Bank's Groundswell project (CIDR, 2022), data by Wang et al. (2024) as well as general information on the settlement development in the region provided by the WSF evolution dataset (Marconcini et al., 2021). As there is a scarcity of data regarding the specific building types present in the region, projected population density is employed as a proxy to estimate whether a building is located in a more rural or more populated area. This allows us to account for the changing development patterns in this region, although it should be noted that all of these datasets display a comparatively low spatial resolution. Additionally, we relate inundation depth to potential damage to buildings, roads, trails, and agricultural areas to quantify the downstream damage potential of a GLOF event. To this end, we rely on previously published depth-damage curves for this region (FEMA, 2009; Huizinga et al., 2017; Chen et al., 2022).

## 4. Methods

Figure 2 summarizes the methodology of this study regarding data acquisition, preprocessing, model setup and the simulations. The following sections provide additional details.

### 4.1 Data preprocessing

The hazard score for each lake (LHL) is calculated by combining the predisposition of glacial lakes to mass-movement impacts (Furian et al., 2021) with permafrost dynamics (Obu et al., 2018; Burke et al., 2023). The permafrost values are derived from two normalized factors: the projected temperature increase (in degrees Celsius) and the change in thaw depth (in meters), each of which is scaled between 0 and 1. The combined permafrost score is obtained by averaging these two normalized values, thereby representing the potential for permafrost destabilization at each point in time. Finally, the overall hazard score is calculated as the weighted average of the lake's mass-movement hazard score and the combined permafrost score, thereby ensuring that the resulting value remains within the range of 0 to 1.

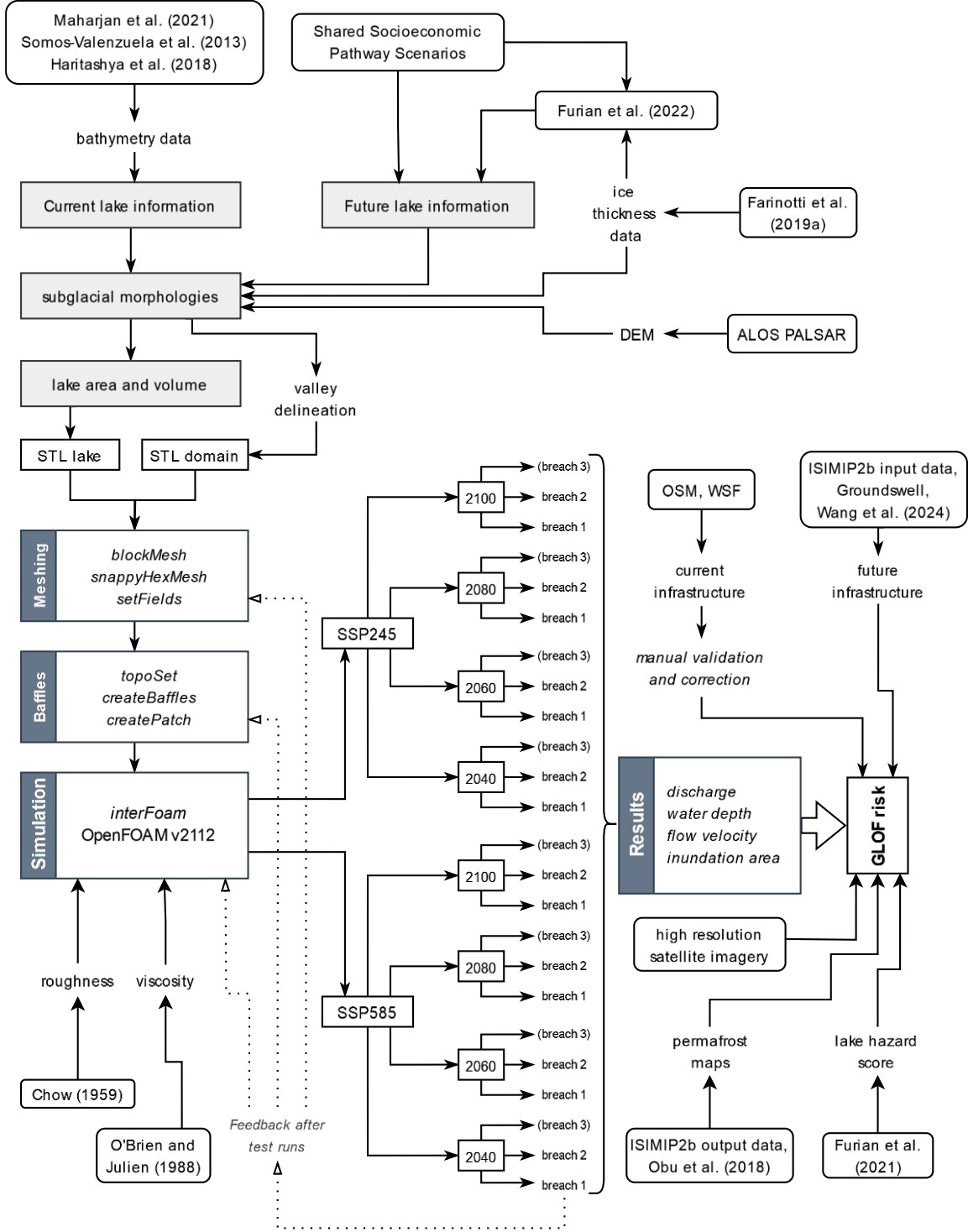

**Figure 2: General overview of the methodology of the study.**

By integrating the data on glacier termini presented by Furian et al. (2022) with the ice-thickness data provided by Farinotti et al. (2019a), we were able to calculate the subglacial morphology for the exposed portion of the glaciers' beds. All bathymetry data and subglacial morphologies were merged with the PALSAR DEM. This allowed us to estimate the potential glacial lake area and volume at various points throughout the 21$^{st}$ century. The locations of probable dam breaches for all lakes were identified using previous studies (Lala et al., 2018; Sattar et al., 2021; Chen et al., 2022) and visual confirmation using high-resolution Maxar satellite imagery in ArcGIS.

Given the inherently unpredictable nature of glacier dam breaches, a scenario-based approach is imperative for more accurately estimating the resulting discharge (Darji et al., 2024). We defined three breach scenarios, each characterized by distinct values for the parameters breach height $h_b$, breach width $B_w$ and moraine failure time $t_f$ (Table 1). While $h_b$ and $B_w$ define the morphology of the breach, $t_f$ indicates the time until the breach incision has reached its full size. Our approach was informed by the methodologies of Froehlich (1995) and Sattar et al. (2021). Given the need for generalization in large-scale 3D simulations using OpenFOAM, these three breach scenarios were applied uniformly across all five study sites. Individually modeling different moraine breaches would not have been feasible, as each geometry must be constructed manually. Moreover, many parameters required to define lake-specific breaches—such as future moraine structure, lake depth, and water levels— are highly uncertain or not readily quantifiable.

To maintain computational feasibility, we did not simulate smaller breach scenarios; the BR1 scenario already represents a moraine incision of 30 meters. This is consistent with several reconstruction studies that model historic GLOFs with moraine breaches between ~25 and ~35 meters (Watanbe and Rothacher, 1996; Somos-Valenzuela and McKinney, 2011; Nie et al., 2020; Mergili et al., 2020; Zheng et al., 2021b). While such a breach would constitute a significant moraine failure, larger events have been documented. Accordingly, BR1 is referred to as a "medium" scenario in this study. The second scenario, BR2, categorized as "high" in this study, reflects the upper range of observed breach dimensions, such as those reported at South Lhonak Lake during the Sikkim flood of 2023 (Sattar et al., 2025). Finally, the third scenario (BR3) represents a theoretical breach of "extreme" magnitude. However, following initial trial runs, we excluded BR3 from the main analysis due to the unrealistically large flood volumes it produced. Although similar breach dimensions have been used in other studies (e.g., Sattar et al., 2021; Mandal et al., 2025), we judged such scenarios to be extraordinarily large. Consequently, the main analysis is based on BR1 and BR2, except at Imja Tsho, where only BR1 was applied due to the moraine's lower maximum height. BR3 results are presented in Tables S1–S3 to illustrate potential worst-case outcomes of GLOFs of extreme magnitude.

**Table 1: Summary of the three breach scenarios.**

| Scenarios | $h_b$ (m) | $B_w$ (m) | $t_f$ (s) |
|---|---|---|---|
| BR1 (medium) | 30 | 140 | 3600 |
| BR2 (high) | 60 | 170 | 2700 |
| BR3 (extreme) | 90 | 200 | 1800 |

Subsequently, the DEM of the study area was modified to create one discrete DEM for each breach scenario. Using GIS software, breach channels through the moraine dams were delineated along the previously determined paths. For each lake, breach channels were created to match the width and height mentioned above. The actual breach modeling in OpenFOAM is described in more detail in Sect. 4.2. From these scenario DEMs, three-dimensional stereolithographic surfaces (STL) were generated, representing the valley downstream of each glacial lake.

Information on built-up area is crucial for demographic studies and crisis response planning, but existing OSM annotations are often insufficient, especially in very rural areas. The main issues include misalignment with updated imagery, incorrect or outdated annotations, and missing annotations for newer buildings (Vargas-Muñoz et al., 2019). We have updated the existing OSM data on buildings along the potential GLOF paths using visual confirmation with Maxar satellite imagery in ArcGIS. Where necessary, the polygons delineating agricultural areas were also modified. To assess the damage that a given inundation depth could cause to these infrastructures and areas, we use the depth-damage curves. Depending on the degree of

damage, we assign each structure to one of three categories: 1–10%, 11– 50%, and 50– 100%, i.e., "slight", "moderate" and "substantial" damage (FEMA, 2009). For the roads, trails, and buildings, information on their characteristics is too sparse to attempt further damage quantification. However, for the agricultural areas, we use information on the average cost of inundated agricultural land (per m$^2$) in this region provided by the Joint Research Centre (JRC) of the European Union (Huizinga et al.,

2017). The damage to crops depends not only on water depth but also the duration of the flooding, however, this is more of a concern in the lowlands due to poor drainage systems (Chen et al., 2022).

## 4.2 Model setup

OpenFOAM (Open Field Operation and Manipulation) is an open-source computational fluid dynamics (CFD) software that provides a versatile framework for modeling complex fluid flow problems, including dam break scenarios. The software is

structured around a modular system of C++ libraries, which allows for extensive customization and flexibility in defining both solvers and physical models (OpenCFD, 2024). We selected the two-phase solver *interFoam* designed to use the Volume of Fluid (VOF) method to handle immiscible, incompressible two-phase flows, which in our case (and typically) involve air and water (Brackbill et al., 1992; Deshpande et al., 2012). OpenFOAM is widely used in CFD and other fields requiring complex simulations of physical phenomena (Chen et al., 2014). Given the availability of comprehensive documentation detailing the

fundamental equations and numerical methods of OpenFOAM, we focus here on the workflow specific to our study and refer readers to the official OpenFOAM manual for further theoretical background (OpenCFD, 2024).

Prior to defining the simulation parameters, the input geometries (i.e., the STL files of the various breach scenarios) are transformed into 3D meshes, comprising each lake and its surrounding slopes, as well as the downstream valley floor up to a length of more than 100 kilometers. To enable the simulation of the two phases (water and air) and their interaction, the

internal mesh extends over 500 meters into the atmosphere above the valley floor. The meshes generated via *snappyHexMesh* by OpenFOAM are composed of hexahedral cells, which are supplemented with additional polyhedral cells near complex geometries to enhance mesh conformity. The generated unstructured grid has a horizontal grid resolution of approx. 20 meters near the surface. This is a good tradeoff between computational costs and accuracy. For the internal and surface meshes near the lake, the breach area and other specific points of interest, we increased the resolution to approximately 10 meters.

Subsequently, a particular water body needs to be delineated within the mesh, i.e., the glacial lake in its spatial extent at each designated point in time. To investigate the changing characteristics of GLOFs during the 21$^{st}$ century, we use the projected extent of glacial lakes for 2040, 2060, 2080, and 2100, based on data by Furian et al. (2022). The current extent of the Imja Tsho, Lower Barun and Tsho Rolpa lakes was determined from Maxar satellite imagery in ArcGIS. The two lakes Bhote and Ngojumba Tsho have not yet formed but are expected to do so in the near future. The dataset by Furian et al. (2022)
also provides projected future areas for these lakes. During the initialization of the model, all grid cells within the lakes were designated as water using OpenFOAM's *setFields* utility. For each lake, a multitude of STL files are generated, with each one representing the lake's extent for one of the numerous possible combinations of SSP scenario and time step.

A breach channel was added to the DEM, incising the moraine, to ensure an accurate cross-section of the dam breach. In many studies (e.g., Zheng et al., 2021a; Yu et al., 2020; Azeez et al., 2020; Jiang et al., 2023), the assumption is made for
the sake of simplicity that a previously defined part of the moraine suddenly disappears, thereby creating an extremely sudden GLOF. In reality, however, the water requires a period of time to deepen the incision into the moraine material and create a substantial discharge. In the present study, a more realistic scenario and hydrograph, which mimics the deepening of the moraine breach, is achieved by incorporating a water permeability control structure—a baffle—into the spillway. The baffle opens non-linearly following a sine wave, simulating a slow-forming breach due to increasing discharge velocity and shear stress.
The breach time

$$t_f = 63.2 \sqrt{\frac{V_w}{g h_b^2}} \qquad (1)$$

is a function of the lake volume $V_w$, the breach height as $h_b$ with g as the gravitational acceleration. The equation is based on Froehlich (1995; 2008), which is among the most commonly employed empirical methods for earthen dam failures (Wahl, 2004). Equation (1) assumes that a breach will always extend to the base of the dam or moraine, which is an implausible
scenario in the context of GLOF modeling. Instead, the potential discharged volume would need to be calculated in advance. However, during the model calibration process, it was determined that minor alterations in the opening time following the equation by Froehlich (2008) have a nearly inconsequential impact on the downstream dynamics of the GLOF. The discrepancies in water velocity, depth, and inundation area were found to be insignificant (see Fig. S1). Accordingly, to reduce the computational time of the simulations, we decided not to recalculate the potential volume, the breach characteristics and baffle

opening times for each simulation run. Instead, we define opening intervals that depend on the magnitude of the breach. We approximate following previous studies (Froehlich, 2008; Lee, 2019; Sattar et al., 2021), which show that the maximum discharge is reached more rapidly as the breach size increases. Accordingly, the baffles in our simulations open at a faster rate in the event of a larger moraine breach (see also Table 1).

It is important to note that GLOFs are typically high-viscosity flows, due to the entrainment of debris and unconsolidated material from moraines and channel beds (Westoby et al., 2014; Meyrat et al., 2024). Nevertheless, many previous studies have opted to simulate GLOFs as clear water to reduce computational complexity (e.g., Sattar et al., 2021; Zheng et al., 2021a; Pandey et al., 2022; Rinzin et al., 2023). This simplification can lead to inaccurate estimates of run-out distance or velocity, as sediment-laden flows have the potential to travel over greater distances due to the increased momentum (Iverson, 1997). In this study, we therefore simulate the GLOF events as hyperconcentrated flows, a widely documented form of outburst flood (Kershaw et al., 2005; Westoby et al., 2014).

While incorporating an explicit sediment transport or a multiphase model could improve process detail even further, it would significantly increase computational demands—especially given the already long runtimes of high-resolution two-phase simulations. Even with HPC resources, it is not feasible to resolve GLOF processes ranging from the transport of fine sediment to the movement of large boulders within a single simulation framework of this scale. We therefore model outburst floods within a two-phase framework, which allows us to capture the essential dynamics while maintaining computational feasibility. In our simulations, the dynamic viscosity

$$\eta = \alpha_1 e^{\beta_1 C_v} \qquad (2)$$

is increased to obtain more realistic GLOF velocity and run-out distance (O'Brien and Julien, 1988). The $\eta$ in Eq. (2) increases exponentially with the volumetric concentration of fine sediments $C_v$. The values for the empirical coefficients $\alpha_1$ and $\beta_1$ are derived from the work of O'Brien and Julien (1988). A value of 0.34 for the sediment concentration $C_v$ is consistent with the findings of Meyrat et al. (2024) and facilitates the modeling of a hyperconcentrated flow, which is characteristic of a GLOF. Using these constants, the viscosity is 5.33 mPa s.

We employ a constant surface roughness value across the computational domain given the scarcity of high-resolution surface roughness data—especially for projecting into future conditions. With this approach, we follow numerous previous GLOF simulations studies (e.g., Larocque et al., 2013; Westoby et al., 2015; Azeez et al., 2020; Majeed et al., 2021; Yang et al., 2023). While the terrain roughness may vary between mountainous and more lowland river reaches, implementing spatially variable roughness values would require splitting the computational mesh into multiple patches or implementing customized boundary conditions, both of which would significantly increase computational time. The constant surface roughness

$$k_s[m] \approx (26n)^6 \qquad (3)$$

we derived from the Manning n value, following the study by Marriot and Jayaratne (2010). In this case, $n$ was set to 0.05 representing a reasonable average for mountain streams in accordance with the existing literature (Chow, 1959; Shrestha et al., 2007). Accordingly, Eq. (3) provides the surface roughness of $k_s = 4.82$.

## 5. Results

In this study, a total of 99 GLOFs were simulated, comprising three from Tsho Rolpa and 24 from each of the remaining four
lakes. Given this multi-scenario nature, it is not feasible to present the results of each individual simulation run in exhaustive detail. Instead, a summary of the results for each run is provided in the supplementary materials (Table S2). For each of the simulation runs, this table presents a comprehensive overview of the inundated areas and the mean water depth (differentiated by residential, natural, and agricultural areas), as well as the number of buildings affected by the flood with their mean flood depth, as well as the length of inundated roads. The following section gives an overview of the minimum and maximum
inundation, which is used to illustrate the range of possible GLOF magnitudes and their impact on the population.

### 5.1 Overview

Figure 3 provides a first overview of the maximum reach of the floods, which is the result of assuming a GLOF in 2100 under the highest SSP scenario with the moraine suffering a BR2 failure. The white circles in this figure indicate the points at which cross-sections through the flood path are utilized to obtain more precise information regarding the properties of the GLOF. For
these cross-sections, locations were selected that had either agricultural areas or infrastructure, e.g., residential areas, bridges, and HPP, in close proximity to the river, where inundation was expected to have a significant impact on local society. As illustrated in Fig. 3, all GLOFs originate the in sparsely populated high-mountain areas, however, depending on the lake, the rivers (and consequently the flood waves) reach the more densely populated areas after ~30–50 km. Simulated GLOFs from every scenario have the potential to reach these areas, and four out of five lakes could, in the high scenarios, potentially produce
massive GLOFs with a range of more than 100 km, covering an elevation difference between 4,000 and 4,600 meters. At the three central lakes (Bhote, Ngojumba and Imja Tsho), important access routes to Mount Everest would be in the flooded area, as well as several settlements that provide infrastructure for trekkers and tourists. Furthermore, each lake has the potential to reach one or more hydroelectric power stations and their associated infrastructure in case of a GLOF.

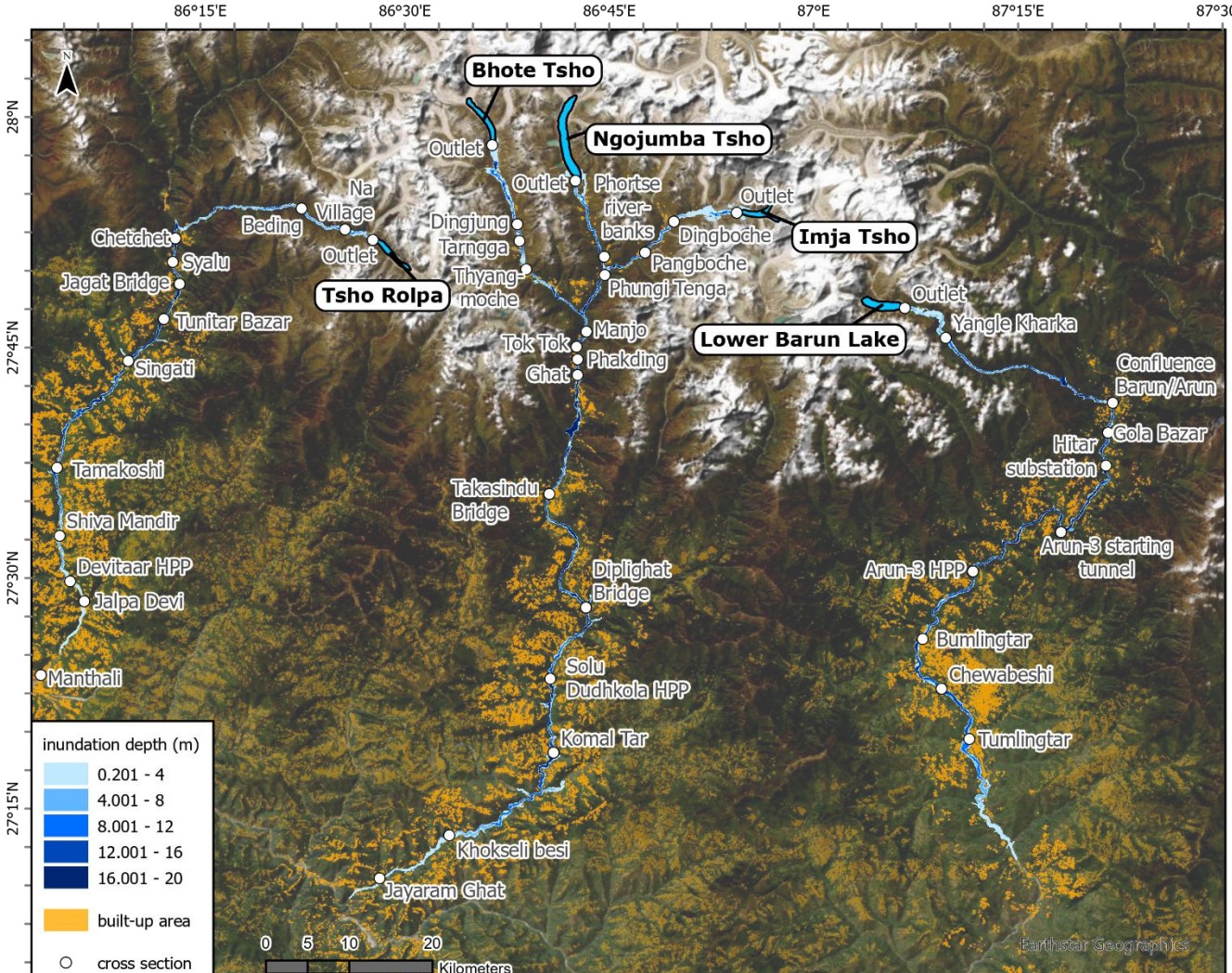

Figure 3: Overview of the maximum possible extent of simulated GLOFs from the five lakes. For Tsho Rolpa, this signifies a GLOF in 2040 due its reduced potential for future growth (SSP2, BR2). For the remaining four lakes, the GLOFs are simulated in 2100 (SSP5, BR2). The built-up area is indicated by orange areas (© OpenStreetMap), which are highlighted and slightly buffered for better visibility in the vicinity of the flooded areas. White named points show the locations of cross-sections used for further analyses.

Table 2 provides a detailed account of the results of the high-magnitude GLOFs and provides insight into their destructive potential. The mean water velocity ranges between 6.3 (Lower Barun Lake) and 8.1 meters per second (Tsho Rolpa), and all GLOFs cover an elevation difference of more than 4,000 meters. Between approximately 750 and 2,000 buildings are directly affected by floodwaters with a mean inundation depth of 5.6 to 7.7 meters. Agricultural areas are particularly vulnerable, as they are more often located close to the riverbanks. Here, ~3.5 km$^2$ could be inundated to a mean depth of 10 meters in the event of a GLOF from the potentially massive Ngojumba Tsho. Because of the alignment of roadways with rivers in high-


mountain areas, each of these high-magnitude GLOFs has the potential to impact between 80 and 100 kilometers of roads or trails, depending on the region.

**Table 2: Impact summary for simulated GLOF events of the highest magnitude.**

| Lake | Distance | Elevation difference | Mean velocity | Affected buildings | | Affected agricultural area | | Affected roads |
|---|---|---|---|---|---|---|---|---|
| | km | m | m/s | Count | Mean inundation depth | km² | Mean inundation depth | km |
| Tsho Rolpa | 85 | 4,018 | 8.1 | 1,989 | 7.1 m | 2.54 | 5.64 m | 100 |
| Bhote Tsho | 109 | 4,422 | 6.7 | 815 | 5.6 m | 3.52 | 4.73 m | 82 |
| Ngojumba Tsho | 113 | 4,375 | 7.1 | 1,042 | 8.5 m | 3.44 | 10.0 m | 96 |
| Imja Tsho | 113 | 4,604 | 7.0 | 1,047 | 5.9 m | 3.14 | 6.34 m | 86 |
| Lower Barun Lake | 100 | 4,247 | 6.3 | 735 | 7.7 m | 0.85 | 7.64 m | 80 |

Once more, the case of Tsho Rolpa is noteworthy as its discrepancy between the maximum and minimum simulations can be attributed solely to the different breach scenarios. In this instance, the SSP scenario and the year of occurrence have no impact on the lake, as it already reaches its maximum size around 2040. Consequently, the differences between high- and low-magnitude GLOFs are less pronounced at this lake than at the others (Table 3).

**Table 3: Impact summary for simulated GLOF events of the lowest magnitude.**

| Lake | Distance | Elevation difference | Mean velocity | Affected buildings | | Affected agricultural area | | Affected roads |
|---|---|---|---|---|---|---|---|---|
| | km | m | m/s | Count | Mean inundation depth | km² | Mean inundation depth | km |
| Tsho Rolpa | 84 | 4,013 | 8.0 | 1,699 | 4.9 m | 2.23 | 3.58 m | 88 |
| Bhote Tsho | 64 | 3,925 | 8.5 | 359 | 1.2 m | 0.88 | 0.9 m | 37 |
| Ngojumba Tsho | 60 | 3,853 | 7.9 | 350 | 2.6 m | 0.45 | 2.42 m | 30 |
| Imja Tsho | 67 | 4,048 | 8.9 | 532 | 2.3 m | 0.59 | 2.33 m | 40 |
| Lower Barun Lake | 59 | 4,017 | 7.8 | 183 | 3.5 m | 0.35 | 3.27 m | 33 |

The difference between the higher and the lower scenarios is significantly more pronounced for the remaining lakes. By 2040, these lakes have not yet reached the volume they can potentially grow to in 2100. Consequently, in the event of a GLOF, the discharge of water would be significantly reduced. Thus, the extent of the GLOFs is reduced, as is the elevation difference. In contrast to the previously maximum of 2,000 affected buildings, the results suggests that only between 183 and 532 structures would be reached by a low-magnitude GLOF. The reduction in outflow volume also leads to a notable decrease in inundation

depths, now ranging between 1.2 and 3.5 meters for the buildings. In addition, the inundation of arable land is significantly reduced. The agricultural area affected by any GLOF of this magnitude does not exceed 1 km² but only reaches between 0.35 and 0.88 km². The average water depth in these areas is between 0.9 and 3.3 meters, depending on the lake. The data also

indicates that the mean velocity of these low-magnitude GLOFs is slightly higher than that observed in high-magnitude scenarios. This can be attributed to the fact that the GLOFs are flowing through steeply inclined terrain for most of their course, whereas GLOFs with a greater discharge volume can reach into the foothills, where they flow at a slower rate due to the flatter topography. This results in a reduction in the average velocity.

The remaining scenarios between the highest and the lowest are not discussed in detail, but in general SSP5 indicates that the glaciers have melted further, resulting in larger lakes compared to SSP2. This means that a greater volume of water is available, which may lead to an increased range of potential GLOFs. As the modeling progresses into the 21$^{st}$ century, glacial lakes continue to expand (except for Tsho Rolpa), which also contributes to larger GLOFs in later years. The same principle applies to breach scenarios: the larger the moraine breach, the greater the volume of water discharged, leading to longer run-outs of potential GLOF events, higher inundation depths and increased infrastructure damage. The volume and characteristics of the moraine breach are the most important factors in determining the potential GLOF risk. Regression analyses indicate that a substantial proportion of the observed variance in the data can be attributed to the year of simulation and the breach, with the SSP scenario exerting a limited influence. The influence of these three parameters on the results varies between the lakes and is illustrated in Table 4.

**Table 4: The results of the regression analyses for the three independent variables indicate that the year and the breach scenario account for most of the variance in the data, while the SSP scenario has a minor impact. Given the limited number of simulations conducted for Tsho Rolpa, it has been excluded from this analysis.**

| Lake | SSP | | | Year | | | Breach Scenario | | |
|---|---|---|---|---|---|---|---|---|---|
| | Min R² | Max R² | Mean R² | Min R² | Max R² | Mean R² | Min R² | Max R² | Mean R² |
| Bhote Tsho | 0.0007 | 0.10 | 0.042 | 0.22 | 0.65 | 0.46 | 0.13 | 0.56 | 0.32 |
| Ngojumba Tsho | 0.001 | 0.68 | 0.17 | 0.007 | 0.60 | 0.27 | 0.16 | 0.74 | 0.39 |
| Imja Tsho | 0.005 | 0.50 | 0.13 | 0.01 | 0.47 | 0.15 | 0.31 | 0.88 | 0.62 |
| Lower Barun Lake | 0.006 | 0.20 | 0.10 | 2.0e-06 | 0.39 | 0.21 | 0.11 | 0.92 | 0.55 |

## 5.2 Lake hazard scores

In order to approximate the hazard of mass-movement impacts at each lake, we calculate adaptive lake hazard scores (LHS) for each lake, depending on the hazard of material detaching from the surrounding mountainsides (Furian et al., 2021) as well as the increasing instability of the upper soil due to permafrost loss (Obu et al., 2018; Burke et al., 2023). The resulting LHS are detailed in Table 5. These scores range between 0 and 1, with higher values indicating a greater hazard. Unfortunately, information on different levels of permafrost degradation for the surroundings of individual glaciers is not available. Given the low resolution of the ISIMIP2b data, the permafrost hazard in this study is represented at the regional level and cannot be calculated on an individual basis for each glacial lake. Nevertheless, the lake hazard score serves as a useful proxy for a lake's

susceptibility to mass-movement impacts from surrounding mountain slopes, taking into account the changing surface stability

throughout the 21st century.

**Table 5: Adaptive GLOF hazard scores for each glacial lake. Values are calculated using previously published slope hazards scores and data on permafrost development in the 21st century.**

| Lake | Lake Hazard Score (LHS) | | | | | | | |
|---|---|---|---|---|---|---|---|---|
| | SSP2 | | | | SSP5 | | | |
| | 2040 | 2060 | 2080 | 2100 | 2040 | 2060 | 2080 | 2100 |
| Tsho Rolpa | 0.23 | 0.26 | 0.23 | 0.23 | 0.26 | 0.39 | 0.51 | 0.52 |
| Bhote Tsho | 0.21 | 0.24 | 0.21 | 0.20 | 0.24 | 0.36 | 0.48 | 0.50 |
| Ngojumba Tsho | 0.26 | 0.28 | 0.26 | 0.25 | 0.28 | 0.41 | 0.53 | 0.55 |
| Imja Tsho | 0.33 | 0.35 | 0.33 | 0.32 | 0.35 | 0.48 | 0.60 | 0.61 |
| Lower Barun Lake | 0.32 | 0.34 | 0.32 | 0.31 | 0.34 | 0.47 | 0.59 | 0.61 |

Contrary to the relatively limited impact of the SSP scenarios on GLOF characteristics, they play a more significant role in

assessing hazard of GLOFs triggered by displacement waves after mass impacts. Table 5 shows the influence of the changing

permafrost on this hazard. Under the SSP2 scenario, the values for all five lakes remain largely unchanged throughout the 21st

century. This is due to the very slow decrease in permafrost until a turning point mid-century, which is followed by a period

of expansion. Consequently, the LHS exhibit minimal variation across all five lakes. In contrast, under SSP5, permafrost is

projected to disappear entirely in this region by the end of the century (Burke et al., 2023). Accordingly, there is a continuous

increase in the LHS from 2040 to 2080. After this point, the scores increase again at a slower rate, reflecting the potential

complete loss of permafrost in this region. In conclusion, it appears that under SSP2, the impact hazard will reach a weak

maximum mid-century and subsequently decrease, though remaining relatively constant. Considering the SSP5 scenario, the

impact hazard can be observed to increase constantly throughout the century. This tendency is more discernible between 2040

and 2080, after which point the growth rate decreases. Nevertheless, towards the end of the century, the LHS are more than

twice as high as under the SSP2.

**5.3 Outburst volume**

A comprehensive examination of all model runs provides insight into the potential volume of water that could be released

during GLOF events, depending on the selected SSP scenario and the specific breach scenario. Figure 4 illustrates the projected

changes in absolute lake volume (black lines) and potential GLOF volume for the 21st century. The latter is indicated by colored

lines with each color representing a breach scenario. The dotted lines illustrate the SSP2, whereas the solid lines show the

SSP5. Additional details on the changing outburst volume in all scenarios can be found in Table S3.

The previously indicated differences between the lakes are also evident in this section. At Bhote Tsho, the rate of

expansion of the lake accelerates only around 2060, then the lake proceeds to quadruple its volume until 2100, when it reaches

a volume of $112\times10^6$ m$^3$ (SSP5). The potential magnitude of a GLOF is significantly influenced by the breach scenarios. BR1

has the potential to drain 33–45% of the lake's volume in a given year, while BR2 could see a release of 62–76%. The SSP scenarios exert a comparatively minor yet discernible influence, particularly around 2080, when the potential outburst volume for all breach scenarios is nearly doubled under SSP5.

The overall volume of Imja Tsho is approximately double that of Bhote Tsho. However, the lake appears to reach its maximum level before the end of the century. The SSP scenarios have a negligible impact on the lake's growth, which appears to be largely decoupled from climate (Furian et al., 2022). The breach scenarios, however, have a considerable

impact on the potential outflow volume. In the BR1 scenario, it is estimated that 40% of Imja Tsho could drain and in BR2, this figure rises to nearly 70%. A similar pattern is observed at Lower Barun Lake, where the data indicate that the lake continues to expand until reaching its maximum size towards the end of the century. The breach scenarios exert a

strong influence on the potential outflow volume. BR1 would allow for a drainage of approximately 45%, BR2 for approximately 80%. The SSP scenarios demonstrate a negligible impact, comparable to that observed at Imja Tsho.

      Glacier modeling indicates that Ngojumba Tsho may expand to

become a substantial glacial lake with a volume of $724 \times 10^6$ m$^3$ by 2100 under the SSP5 scenario (Furian et al., 2022). The potential outflow volume is not significantly affected by the SSP scenarios until 2060. Subsequently, however, the SSP scenarios exert an increasingly high influence, in conjunction with the breach scenarios. Given the considerable dimen-

sions of the lake, the BR1 scenario would result in a drainage of 18–20% and the BR2 scenario in 35–46%. It is possible that these figures will increase further, given that the lake is expected to expand further in the 22$^{nd}$ century. In contrast, the growth of Tsho Rolpa is projected to be complete by 2040, regardless of potential climatic changes. It is solely the selected

breach scenario that exerts an influence on the magnitude of the GLOF.

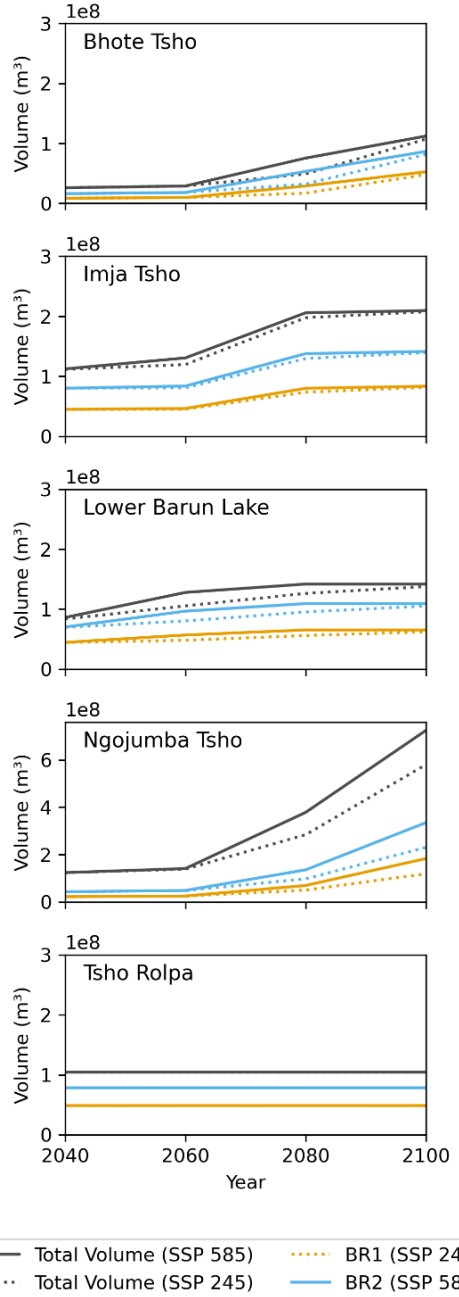

**Figure 4: Volume of water available during the simulated GLOF events during the 21st century. Black lines indicate the total lake volume, while the colored lines show the outflow volume for the breach scenarios. Dotted lines represent SSP2, while the solid lines indicate SSP5. Note the different Y-axis scale for Ngojumba Tsho.**

In the case of BR1, it is estimated that up to 45% of the total volume of the lake ($104×10^6$ m$^3$) could be released. In the case of BR2, this figure rises to 75%.

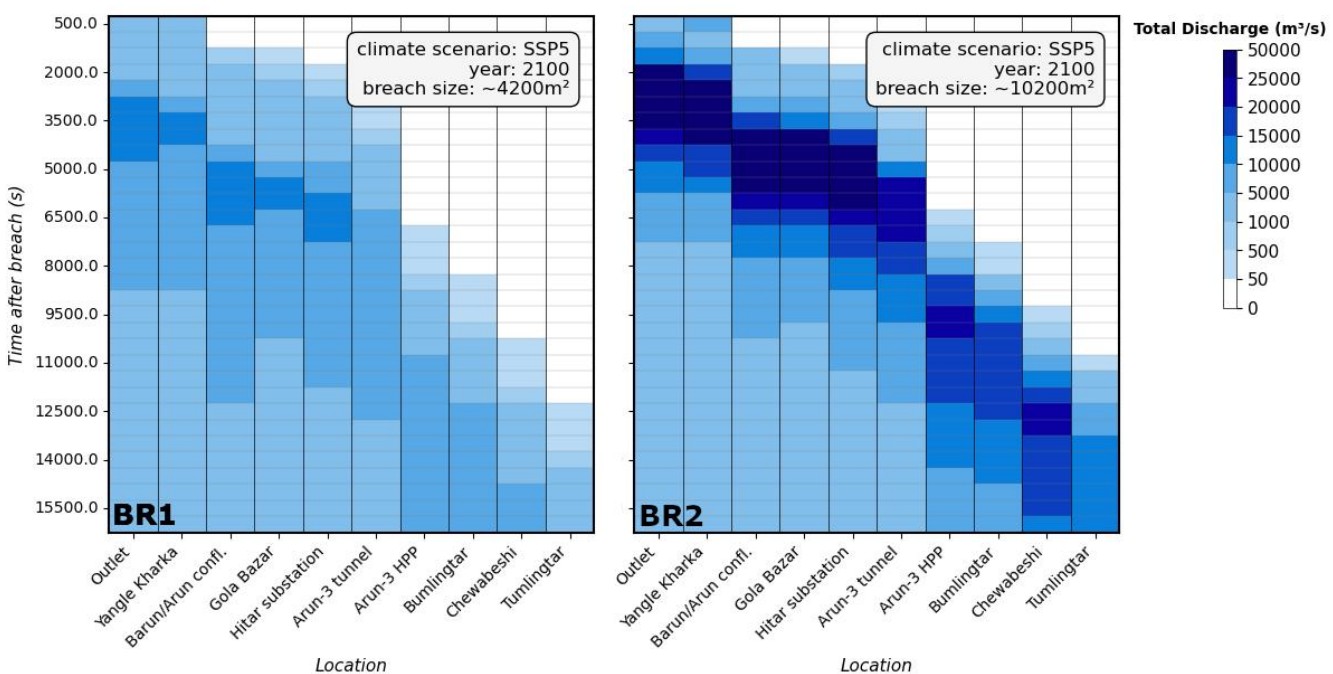

**Figure 5: Comparison of GLOFs from two moraine breach scenarios at the Lower Barun Lake in 2100. Shades of blue indicate the discharge at the downstream locations. In the higher breach scenarios, the discharge is increased, while the travel time of the GLOF is reduced.**

## 5.4 Spatiotemporal discharge analyses

The cross-sections (see Fig. 3) allow to investigate the changes in discharge along the GLOF path, as well as the timing. Figure 5 depicts the projected consequences of a GLOF at the Lower Barun Lake in 2100, modeled under the SSP5 scenario. The lower BR1 scenario results in a GLOF with a maximum discharge of 13,603 m$^3$/s after 58 minutes. The small settlement Yangle Kharka is situated approximately seven kilometers downstream of the lake and will be affected by the initial wave (2,154 m$^3$/s) after eight minutes. After one hour, the GLOF reaches its maximum discharge of 13,154 m$^3$/s at this location. Further downstream, the construction sites for the Arun-3 HPP are situated within the inundated zones. After 50 minutes, the site of the HPP's starting tunnel experiences an increase in the discharge of the Arun River due to the GLOF, with initially 115 m$^3$/s. In the subsequent hour, the additional discharge at this location reaches its maximum of 8,806 m$^3$/s. The construction site of the powerplant will be affected by the initial wave approximately two hours after the moraine breach, and the full GLOF will reach it after 3.5 hours, resulting in an additional discharge of 8,247 m$^3$/s. The city of Chewabeshi is reached initially after 2.8 hours, with the discharge increasing from 55 m$^3$/s to 6,430 m$^3$/s at the end of the simulation. Data indicate that the maximum

discharge may not be reached at this point. The same is true for Tumlingtar, situated even further downstream, where the GLOF initially hits after 3.3 hours. After 4.4 hours, the discharge increases up to 3,065 $m^3$/s with the potential to increase further.

With the BR2 moraine breach, the maximum discharge at the outlet of the Lower Barun Lake reaches 44,300 $m^3$ after 42 minutes. At Yangle Kharka, the initial impact of the GLOF is a discharge of 5,072 $m^3$/s, which increases to 42,005 $m^3$/s
within 50 minutes. After 40 minutes, the first wave (362 $m^3$/s) reaches the site of the starting tunnel for the Arun-3 HPP. 20 minutes later, the site is impacted by the full GLOF with an additional discharge of 24,617 $m^3$/s. The Arun-3 HPP is initially affected after 100 minutes, with a discharge of 167 $m^3$/s. After 2.6 hours, the full GLOF reaches this site and increases the discharge of the Arun River by 20,751 $m^3$/s. Chewabeshi is impacted after 2.5 hours. In the next hour, the discharge continues to increase until it reaches its maximum of 20,637 $m^3$/s. In Tumlingtar, the GLOF can be measured after 2.9 hours, with the
maximum discharge (13,231 $m^3$/s) occurring after 4.3 hours.

## 6. Discussion

### 6.1 Uncertainties and model limitations

An analysis of the characteristics of future GLOFs must account for a number of uncertainties inherent in the data and methodology used. The ALOS PALSAR DEM performs comparatively well in mountainous regions (Xu et al., 2024); however, it
still exhibits a root mean square error (RMSE) of approximately 9 meters (Chai et al., 2022). As a result of this inherent uncertainty, there may be discrepancies between the findings of this study and those of other studies regarding the lake surface elevation or the valley topography. In the modeling of glacier retreat and the resulting formation of glacial lakes, it is essential to also consider the uncertainties associated with the chosen models and climate scenarios, which are discussed in detail by Furian et al. (2022). As their data serve as the foundation for modeling the future volume of the lakes, any uncertainties inherent
to them will have an impact on the results.

In this study, we did not model the physical incision into the moraine material; instead, we opted for the use of baffles to simulate this process (see Sect. 4.2) to reduce the computational demand. This approach ensures the necessary comparability between scenarios. However, the occurrence of moraine breaches depends upon a multitude of factors, including the composition of the moraine material, the internal temperature, the presence of an ice core, and numerous other variables. In the
absence of data regarding the future characteristics and material composition of moraines, the selected breach scenarios delineate a range of potential moraine breach events.

In contrast to other studies, we implemented a higher viscosity to simulate the GLOFs as hyperconcentrated flows rather than as clear-water flows. While this approach simplifies the calculations, it also precludes the possibility of the GLOF changing its characteristics during its course. A typical GLOF initially may start as a mudflow, it usually evolves into a

hyperconcentrated flow as it incorporates eroded solid debris (Westoby et al., 2014). When sufficient material is present, it can further transform into a debris flow, which may subsequently revert to a hyperconcentrated flow as the material accumulates in areas with a gentler slope and the separated fluid flows downstream. While it thus cannot reflect the effects of possible changes in solid content, the simulation of GLOFs as hyperconcentrated flows with a sediment content of ~30% serves as a good approximation (Meyrat et al., 2024).

CFD modeling is not without its own set of uncertainties, mainly regarding the resolution of the mesh, which is, in this case, somewhat coarser than the resolution of the DEM. To place our results in a larger scientific context, we compared our results with previous GLOF simulations at Tsho Rolpa, Lower Barun Lake and Imja Tsho. Despite the different modeling approaches, our results generally align with these studies. A visual comparison with the results of Sattar et al. (2021) reveals similar flood extent and inundation patterns at Lower Barun Lake, although our simulations produce slightly higher flow

velocities and inundation depths at certain downstream locations. Mandal et al. (2025) report comparable maximum inundation depths of approximately 20 meters for a 50-meter breach, which aligns well with the results from our 60-meter BR2 scenario. Reported flow velocities ranging from 3 to 10 m/s are also in good agreement with our estimates, which fall between 3 and 8 m/s.

At Tsho Rolpa, Chen et al. (2022) simulated a GLOF from a 30-meter moraine breach reaching as far as Manthali— closely matching the run-out distance of our BR1 scenario. However, the higher discharges in our model lead to higher velocities. Several other studies report discharge and inundation depths consistent with our findings: Shrestha et al. (2012) estimated a peak discharge of 90,000 $m^3 s^{-1}$ in their highest scenario, compared to the 81,000 $m^3 s^{-1}$ in our simulation. Chen et al. (2025) reported peak discharges between 13,000 and 15,000 $m^3 s^{-1}$, which aligns well with our value of approximately 12,300 $m^3 s^{-1}$. Both discharge and inundation depth estimates are in good agreement with the study by Kayastha and Maskey (2024), which

simulated GLOFs from moraine breaches with a width of 20 meters and 40 meters. At Imja Tsho, Somos-Valenzuela et al. (2015) compared different lake lowering scenarios, which could account for their lower discharge and GLOF run-out estimations. Chen et al. (2025) estimated a mean discharge at Imja Tsho of 15,000 $m^3 s^{-1}$, which aligns well with the 11,800 $m^3 s^{-1}$ in our model.

These comparisons indicate that while 3D modeling of GLOFs generally aligns with results from 2D simulations, it reveals notable distinctions in the specific flow characteristics. These differences, though subtle in some cases, are more substantial regarding inundation depth and flood reach. Although the computational demands are higher, several studies have demonstrated that the results of 2D flood modeling can be further refined by the application of a 3D modeling approach (e.g., Akgun et al., 2023; Dellinger et al., 2024; Lee et al., 2024). It can thus be assumed that our results accurately represent the inundated areas and affected infrastructure due to the more precise representation of complex flow patterns in high-mountain

terrain and the more realistic viscosity.

## 6.2 GLOF hazard assessment

The adaptive lake hazard scores (LHS) indicate how the hazard of GLOFs caused by detached material impacting glacial lakes will change in the 21st century under different climate scenarios. As indicated by ISIMIP2b for both climate scenarios, the loss of permafrost in this portion of the Himalayas remains comparatively limited in comparison to other regions. Regarding the five lakes, the hazard of impact-induced GLOFs is relatively low under SSP2 (LHS = 0.22-0.32). Under SSP5, the loss of permafrost is clearly apparent, as evidenced by higher LHS values between 0.52 and 0.61. This means that a higher climate scenario is associated with an increased hazard of landslides or rock falls, which in turn would result in an increased GLOF hazard at some locations.

At Bhote Tsho, glacier modeling suggests a potential lake formation around 2030 with an initially slow growth that accelerates in subsequent decades (Furian et al., 2022). Although the hazard of impact-triggered GLOFs is the lowest among the five lakes due to the relatively stable adjacent slopes, a potential moraine failure could still result in a significant GLOF. Similarly, the formation of Ngojumba Tsho remains uncertain. However, the maximum volume could exceed $700\times10^6$ m$^3$, especially under the SSP5 scenario, which would make it one of the largest lakes in the region. The GLOF hazard increases under SSP5 due to the accelerated permafrost loss and subsequent potential slope destabilization. Additionally, there is the possibility of a second proglacial lake forming at Gaunar Glacier, which joins with Ngojumba's glacier tongue. A GLOF from this lake could trigger significant displacement waves when impacting Ngojumba Tsho, which could result in a moraine breach at the latter.

The fastest growing of the five lakes is the Lower Barun (Haritashya et al., 2018), which has a high hazard of impact-triggered GLOFs due to surrounding steep slopes, a calving glacier, and unstable cliffs. Under SSP5, the risk continues to rise due to the degradation of permafrost, increasing the likelihood of avalanches or rock falls. Such could generate displacement waves of sufficient magnitude to breach the wide moraine due to tsunami-like ramp-ups caused by the rising lake bed. The susceptibility of Imja Tsho to GLOFs is linked to its moraine structure and ongoing lake expansion. The hazard is moderate under SSP2 due to partial permafrost stabilization but significantly higher under SSP5, where the moraine could be breached by displacement waves, especially given its comparatively low freeboard (~10 meters) and weakened ice core (Bajracharya et al., 2020). At Tsho Rolpa, the hazard of impact-triggered GLOFs is high due to the proximity of hanging glaciers and a minimal freeboard of around 5 meters, which increases the likelihood of moraine overtopping (Bajracharya et al., 2020). The potential for a substantial GLOF event is increased under SSP5 due to the rapid permafrost degradation, which could destabilize the surrounding slopes. This is especially problematic if the anticipated second lake forms behind the glacier's northward bend (Furian et al., 2022), which would indicate a cascading GLOF hazard.

## 6.3 Downstream impacts

For our analysis of the GLOF impact, we include the estimated hazard of an impact-triggered GLOF as well as information on the impact downstream. This includes the damage done to inundated structures and areas, the magnitude of the GLOF as well as the response time at each location. We assess the damage potential of selected GLOF scenarios by quantifying the damage to buildings, roads, trails, and agricultural areas. When properly installed and maintained, early warning systems (EWS) can help alert the local communities and, at least in settlements further downstream, help to reduce the impact of a GLOF due to a sufficiently long response time. However, of the five lakes, only Imja Tsho appears to have a functioning EWS (Deepak et al., 2021; Wang et al., 2022a), since the system at Tsho Rolpa was discontinued. The Lower Barun has, to our knowledge, never had an EWS installed, despite its potential GLOF risk.

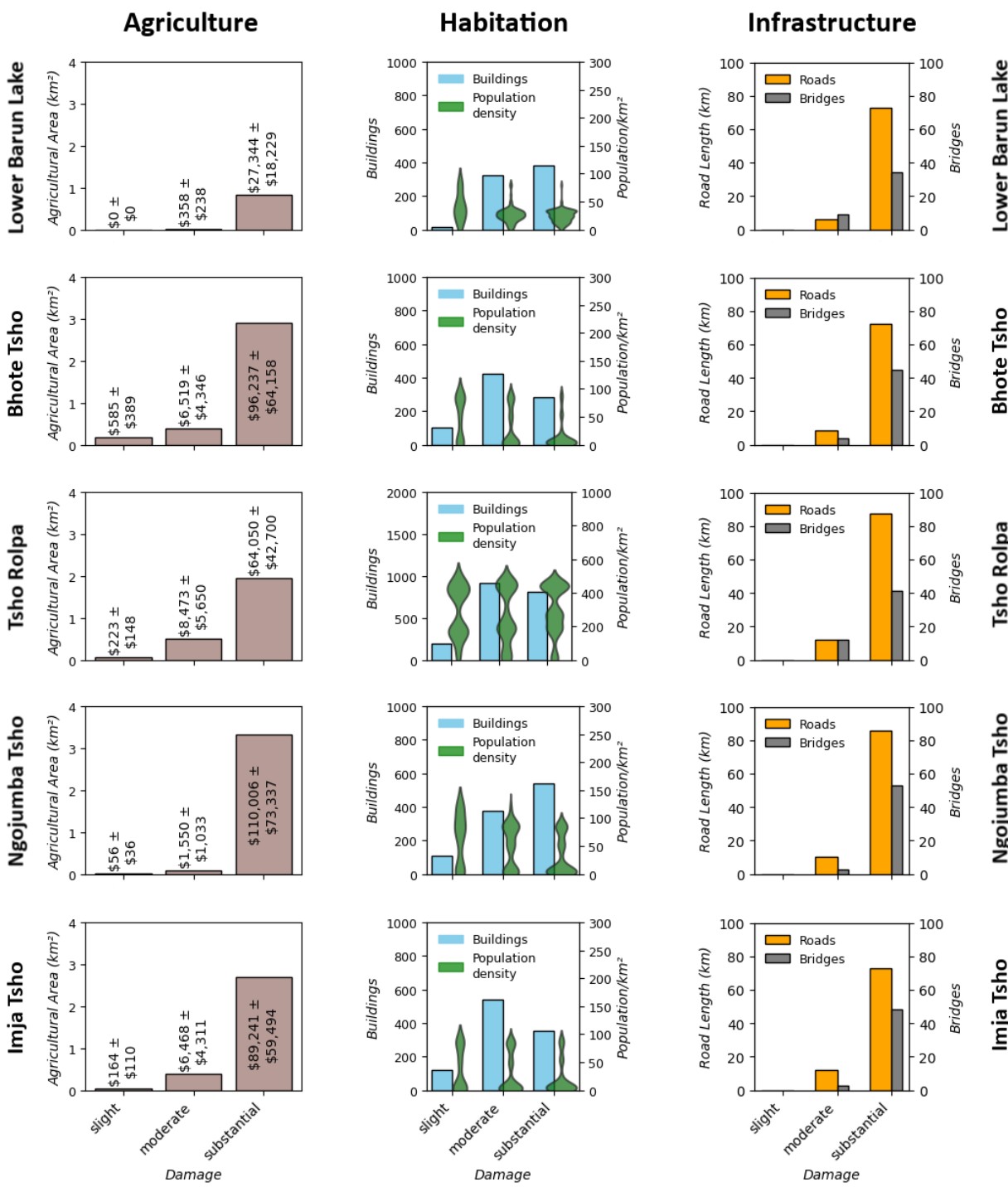

**Figure 6: Downstream impact of a high-magnitude GLOF in 2100 under the SSP5 scenario (2040 and SSP2 for Tsho Rolpa). The potential damage is based on depth-damage curves (Huizinga et al., 2017; Chen et al., 2022). The financial impact is based on lower and upper restoration cost per m² of agricultural land in this region (Huizinga et al., 2017). The population forecasts are provided by Wang et al. (2024). Note the different scale of the Y-axes in the habitation plot for Tsho Rolpa.**

### 6.3.1 Economic impacts

In Fig. 6, the impact of high-magnitude GLOFs on buildings, infrastructure and agricultural land is put into a broader context for each lake. The figure illustrates the impact of a high-magnitude GLOF from each lake, classified into three categories: agriculture, habitation and infrastructure. The X-axes of the individual plots indicate the extent of damage to the respective structure, categorized as slight (1-10%), moderate (11-50%) or substantial (51-100%) damage. A second figure detailing the results of lower-magnitude GLOFs can be found in the supplementary material (Fig. S2).

It is evident that the impact on agricultural land is significant across all lakes. The Lower Barun is an exception, as agriculture is less prevalent in this area and the riverbanks are generally less densely populated. In all scenarios, however, only a limited number of areas are affected by slight flooding, while a significant percentage of the agricultural area is subject to heavy flooding and substantial damage. This illustrates the extent of damage that a GLOF would inflict on agricultural land in the vicinity of the river, due to the considerable inundation depths, high flow velocity and potentially high sediment load. 585 Subsistence farming is of particular significance in these remote regions (Panthi et al., 2016), and GLOFs of this magnitude would have a profound impact on the food supply for the local population. Moreover, the financial investment required to restore the agricultural land would be considerable. The financial burden of restoring agricultural land following a GLOF from Imja Tsho, contingent on the crops grown, ranges from $30,000 to $148,000. This would place a significant financial strain on the communities. Should the Ngojumba Tsho develop in accordance with the simulations by Furian et al. (2022), the resulting 590 damage could be significantly greater with nearly $200,000. It is crucial to acknowledge that these estimates solely represent the costs associated with the restoration of damaged agricultural land. Any additional costs associated with replacing lost crops or livestock are more challenging to quantify and thus fall outside the scope of this study. As it is possible that these additional costs could be more than double our estimates, they would serve to intensify the financial pressure on the mountain communities.

The projected population development for the Everest region is comparable across the surroundings of all GLOF paths. Until 2080, settlements further encroach on the higher mountain areas, accompanied by an increase in population density in the more populated areas, especially along the path of GLOFs from Tsho Rolpa. This phenomenon is more pronounced under SSP2 than under SSP5. From 2080 towards the end of the century, the density decreases slightly in both scenarios. However, in the more rural areas along the potential GLOF paths, the population density slightly decreases throughout the 600 entire century, probably due to migration to the larger settlements (CIDR, 2022; Wang et al., 2024). This suggests that GLOFs are encountering a relatively stable population throughout the century, although the slight decreases indicate reduced economic impacts in later decades.

In the results of our GLOF modeling, the ratio of moderate to substantial building damage is more balanced than the agricultural impact. The majority of GLOFs exhibits a greater proportion of buildings that would be classified as only slightly or

605 moderately damaged rather than substantially damaged. It is notable that any Tsho Rolpa GLOF would affect much more densely settled areas, with population densities exceeding 450 people per square kilometer. This increases the probability that the affected buildings will include critical infrastructure such as hospitals, schools, or HPPs. The other lakes predominantly affect sparsely populated regions. While this indicates that a smaller number of individuals would initially be affected, the infrastructure in these regions is less equipped to provide assistance following a flood. In the scenario in Fig. 6, the provision

of post-disaster assistance is further complicated by the fact that GLOFs of this magnitude would also result in significant damage to roadways and bridges. Aside from these immediate impacts, all simulated GLOFs have the potential to disrupt local tourism by damaging critical infrastructure such as bridges and hiking trails. Disruption of these structures would not only hinder tourist flows but also inflict ripple effects across local economies reliant on trekking services, accommodation, and guiding (Nyaupane and Chhetri, 2009). This vulnerability is a growing concern, especially in the Everest Region, where a

positive correlation has been found between employment in the tourism industry and increased awareness of GLOF risk (Sherpa et al., 2019). These results underscore the need for multidisciplinary local adaptation strategies that integrate both vulnerabilities and economic opportunities linked to tourism within comprehensive GLOF-risk management frameworks (Khadka et al., 2025).

### 6.3.2 Local impacts

To provide a more concrete representation of the results of this study, downstream impacts for each currently existing lake are presented at an exemplary location. Downstream of Lower Barun Lake, several hydropower projects are in operation or under construction, most notably the Arun-3 HPP, which with its 900 MW will be one of the largest HPP in Nepal (Global Energy Monitor, 2024). Located at about 45 km from the lake is the Arun-3 starting tunnel, which is being constructed at a sharp bend in the Arun River (Fig. 7). According to our modeling, this site could already be severely impacted by GLOFs of lower mag-

nitude. Differences between our scenarios mainly concern the flow velocity which increases with larger discharge volumes and/or greater moraine breaches and can assume average values between 7 and 12 m/s. Accordingly, the arrival time of the main flood at the starting tunnel varies between ~1h and 1.75h.

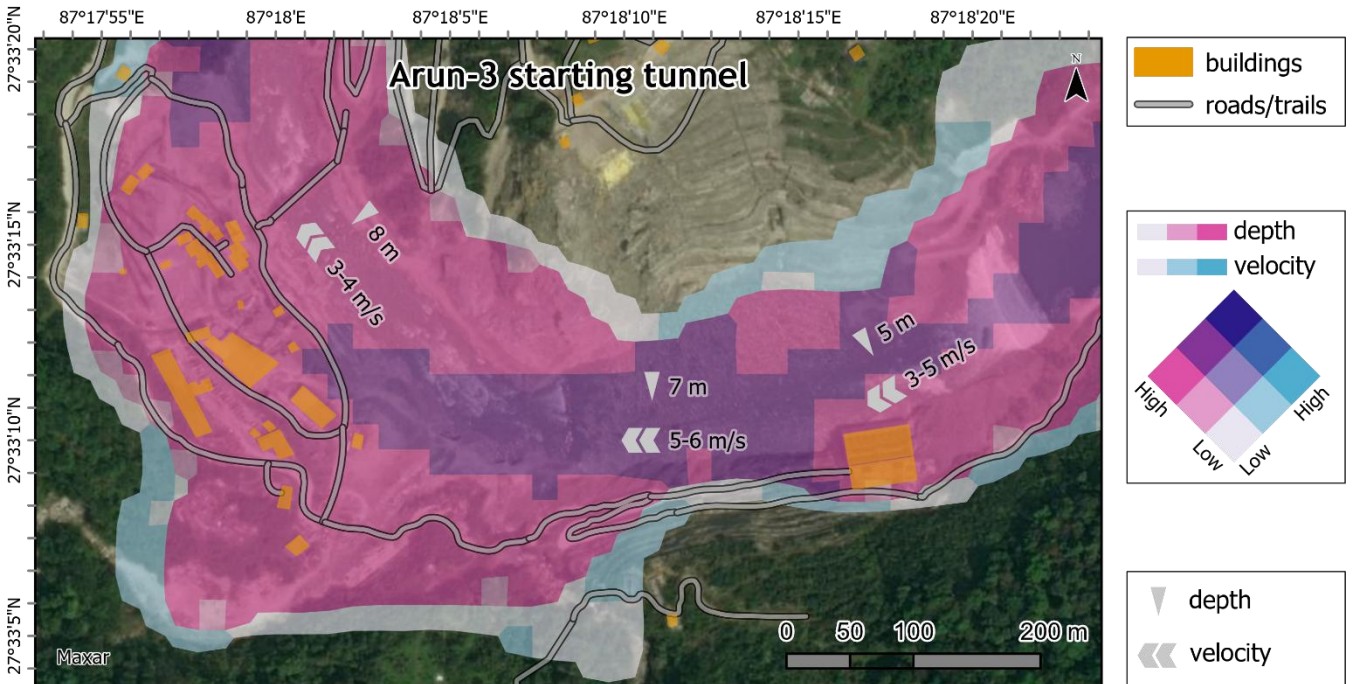

**Figure 7: The construction site of the starting tunnel belonging to the Arun-3 HPP is severely impacted by this simulated low-magnitude GLOF in 2040 (SSP2, BR1). After one hour, the discharge of the Arun River increases by a maximum of 8,806 m³/s and inundates the entire construction site on the southern riverbank as well as the tunnel opening on the northern bank.**

The LHS of the Lower Barun Lake in the year 2040 is 0.32, which indicates a moderate hazard of impact-triggered GLOFs. The degradation of permafrost remains low under the SSP2; however, the combination of the steep relief and the calving glacier provides sufficient opportunity for mass movements to impact the lake. In Fig. 7, the situation at the Arun-3 starting tunnel one hour after the breach occurred is depicted. At this point in time, the main flood wave from a low-magnitude GLOF in 2040 (SSP2 and BR1) would reach the site with a maximum additional discharge of 8,806 m³/s. With a velocity of up to 6 m/s and a depth of more than eight meters, the Arun River would then overflow and inundate the buildings on the southern riverbank. Given that this HPP is of the run-of-the-river type, the absence of a large reservoir prevents the possibility of using it as a means of flood mitigation. Instead, a GLOF of this magnitude would directly impact the HPP's headrace tunnel as well as powerhouses and other structures on the riverbank. Due to the debris and sediments carried by a potential GLOF, the damage could be severe. Therefore, these results indicate the urgent necessity for the installation of GLOF adaptation measures at the HPP as well as the installation of an EWS at Lower Barun Lake.

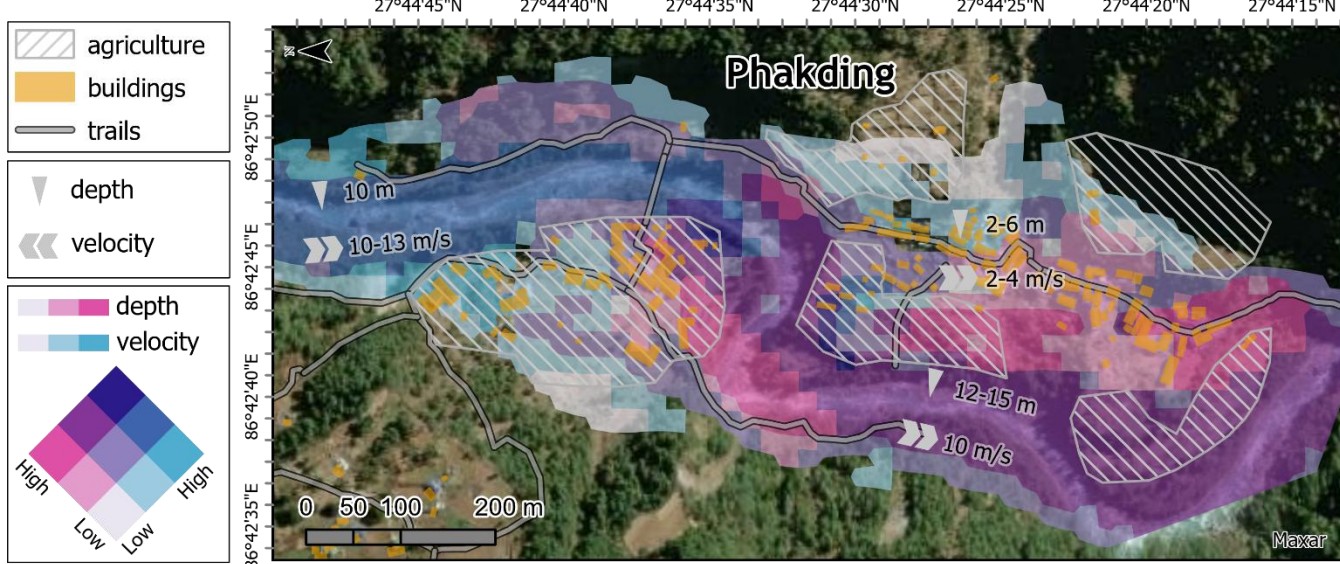

**Figure 8: The village of Phakding is heavily affected by a GLOF of medium magnitude (2080, SSP5, BR1) from Imja Tsho with nearly every building and several agricultural areas at least partly inundated.**

The small village of Phakding is situated on the banks of the Dudkoshi River, approximately 33 km downstream of Imja Tsho. It is a UNESCO World Heritage Site and essential destination along the trekking route to Mount Everest (Mayhew and Bindloss, 2009). The village has the capacity to accommodate approximately 300 tourists in its 25 hotels. However, in the event of a GLOF, this would increase the vulnerability of the village. Should the current trajectory of increasing high-mountain tourism continue, the significance of the village is likely to increase over the coming decades, accompanied by an expansion of tourist accommodation and related infrastructure. In Fig. 8, a GLOF of medium magnitude impacts Phakding in the year 2080 (SSP5, BR1). With an LHS of 0.6 in 2080, Imja Tsho has a high hazard of impact-triggered GLOFs, due to the ongoing lake expansion and the retreating permafrost, which is destabilizing the surrounding slopes. In this scenario, the main flood reaches Phakding ~110 minutes after the breach at Imja Tsho. The maximum velocity and water depth appear to be confined to the river bed due to the elevated riverbanks. However, with a velocity of up to 13 m/s, the flood wave would overtop the riverbanks and flood the village. Our simulations indicate that a GLOF of this magnitude could inundate most of the village, including nearly every building and a significant portion of the agricultural areas, with a depth of up to six meters. Furthermore, the bridge and the trails leading to and from the village are also affected, which would impede the delivery of aid to the village in the aftermath of the GLOF. These findings reinforce the necessity of maintaining the EWS at Imja Tsho, as the ~110-minute window might prove crucial for residents and tourists in Phakding to evacuate the village.

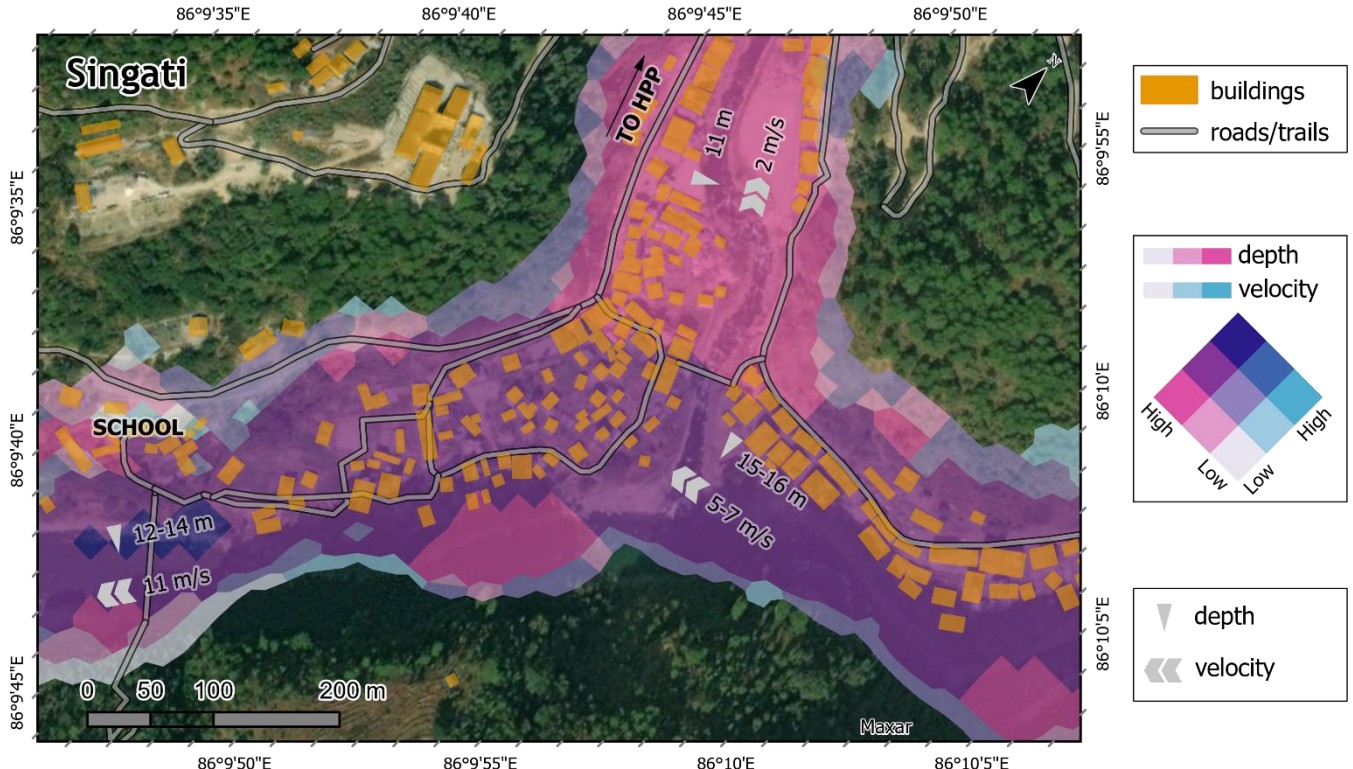

**Figure 9: The village of Singati is severely impacted by this high-magnitude GLOF from Tsho Rolpa (current situation, BR2). Several bridges and buildings area affected, including a boarding school and a small HPP in a tributary river.**

Approximately 37 km downstream of Tsho Rolpa is Singati, a village situated at the confluence of the Singati River and the Tamakoshi River. Tsho Rolpa has almost reached its maximum size already and its hazard of impact-triggered GLOFs is high due to its steep moraine and the small freeboard (see Sect. 6.2). The GLOF hazard will therefore increase over the next decades not due to lake growth, but because of permafrost degradation and a potential cascading hazard due to a potential second lake. The simulated GLOF presented in Fig. 9 depicts a scenario of a high-magnitude GLOF (BR2), which theoretically could occur

at any time. Approximately 1.5 hours after the GLOF is triggered, the main flood wave reaches Singati, resulting in the complete inundation of the village area near the confluence. The destructive potential of the GLOF is considerable, given the depth of the water (up to 16 meters) and the velocity (exceeding 10 m/s). In the southernmost region of the village, where the local boarding school and the bridge spanning the Tamakoshi River are situated, the impact of the GLOF will be particularly pronounced. Due to the considerable depth of the water, the flow will enter the valley of the Singati River, with the potential to

reach and potentially damage the small HPP situated 1 kilometer upstream. Fortunately, the village's hospital is situated at a higher elevation and will not be affected. These findings reinforce the necessity for the reestablishment of the EWS at Tsho Rolpa, given that the elevated GLOF hazard is accompanied by a significant potential for downstream damage.

## 7. Conclusion

This study employed the three-dimensional CFD model OpenFOAM to simulate glacial lake outburst flood events at five glacier lakes in the Everest region of Nepal, with the aim of assessing how GLOF characteristics could evolve throughout the 21$^{st}$ century under various breach scenarios and socioeconomic development pathways. Our findings indicate that in low-magnitude scenarios (2040, SSP2, BR1), the five lakes have the potential to produce GLOFs that inundate between 0.35 and 2.23 km$^2$ of agricultural land, with an average water depth ranging between 0.9 and 3.58 m. The simulated GLOFs travel between 59 and 84 km, impact 30–88 km of roads or trails, and inundate 183 to 1,699 buildings with 1.2 to 4.9 meters of water. These values are significantly exceeded in the higher scenarios. Towards the end of the century, the hazard of an impact-triggered GLOF increases due to the potential for permafrost shrinkage to destabilize slopes and moraines. While the population in rural areas is expected to decline slightly after 2080, the GLOFs now predominantly reach distances exceeding 100 km and affect larger settlements in the foothills. According to our results, 80 to 100 km of roads or trails, 735 to 1989 houses, and between 0.85 and 3.52 km$^2$ of agricultural land could be flooded, with average water depths reaching up to 10 meters. The higher water volumes result in higher flow velocities for high-magnitude GLOFs in some cases, which reduces the warning time for downstream settlements if an EWS is implemented at the lakes.

In three in-depth examples, we delineate the effects of a low-, a medium- and a high-magnitude GLOF on selected locations downstream. For the low-magnitude GLOF (2040, SSP2, BR1), we analyze its impact on the Arun-3 hydropower plant. With flow velocities of up to 6 m/s and water depths of up to eight meters, the headrace tunnel (or, at present, its construction site) would be heavily affected, which could potentially result in significant damage to one of Nepal's largest HPPs. It is evident that GLOF protection facilities are essential and should be implemented during the construction phase. We underline the potential benefits that the population downstream of Lower Barun Lake could gain from the implementation of an EWS.

In an exemplary in-depth analysis of a medium-magnitude GLOF (2080, SSP5, BR1), we demonstrate that even settlements situated on elevated riverbanks, such as Phakding, are susceptible to inundation. While the majority of the water remains within the riverbed (with velocities reaching 13 m/s and depths of up to 15 m), the flooding of buildings and agricultural areas in the village, a significant tourist destination, is almost inevitable. Thus, it is crucial that the EWS at Imja Tsho is continuously maintained to ensure the safety of the village's population.

To illustrate the potential impact of a high-magnitude GLOF, we examine the case of Singati, a village situated downstream of Tsho Rolpa. Our analysis suggests that this village would likely face severe destruction in the event of a GLOF. Most structures situated in close proximity to the river could be submerged by water levels reaching up to 15 meters, with the potential for significant structural damage due to the high velocity of up to 14 m/s. While the hospital would be spared, the

local boarding school is situated within the flood zone. This reinforces the necessity of monitoring Tsho Rolpa and the value of an EWS in providing Singati with approximately 1.5 hours' advance warning of the GLOF's arrival.

Despite the inherent uncertainties of three-dimensional GLOF modeling for the 21$^{st}$ century, our results align well with those of previous publications in this field. However, due to the enhanced precision of 3D flood modeling in high mountains, our results are anticipated to offer valuable insights into the evolution of GLOFs in the 21$^{st}$ century due to the high spatial resolution of our model and the detailed simulation of turbulence, viscosity, and moraine breach time. The information can facilitate more accurate assessments of future GLOF risk in the region, supporting the implementation of essential monitoring

and investigation procedures at potentially hazardous lakes, the establishment of EWS when feasible, and the implementation of additional GLOF adaptation strategies.

**Code availability**

The code for the CFD model, along with detailed documentation and tutorials, is available on the OpenFOAM website (OpenCFD, 2024). The workflow of this study, including instructions and code examples, is accessible via GitHub and ar-

chived on Zenodo (Furian, 2025).

**Supplementary Material**

The Supplementary Material for this article can be found online at: ___.

**Author contributions**

WF and TS developed the idea for this study. WF created the outline and designed the final structure together with TS. Both

TS and WF determined the model parameters and WF performed the CFD simulations with support by TS. WF wrote the code for the subsequent analyses, performed them, evaluated and interpreted the results. TS contributed to the interpretation of the results. WF wrote the first version of the manuscript. Both authors participated in structuring and finalizing the manuscript.

**Competing interests**

The authors declare that they have no conflict of interest.

**Financial support**

WF was funded by a personal grant of the Studienstiftung des Deutschen Volkes. We acknowledge support by the German Research Foundation (DFG) and the Open Access Publication Fund of Humboldt-Universität.

## Acknowledgements

We thank OpenCFD Ltd. for providing and continuing to support the OpenFOAM model. We also thank the OpenFOAM community for the support during the simulations. The research conducted in this study was made possible by several other open-source software tools, such as PyCharm and ParaView, whose development teams we hereby thank. We also thank all researchers and research groups who made data freely available that has been used in this study. Christoph Schneider is thanked for his support during the early stages of this study. We acknowledge the contribution of Anselm Arndt, who, together with TS, provided the possibility to use the HPC at Humboldt-Universität, enabling this comprehensive CFD study. We also thank the two reviewers, Adam Emmer and Binod Diwadi, for their helpful comments and suggestions, which considerably improved the final manuscript.

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
