# Peer review of "Assessing economic impacts of future GLOFs in Nepal's Everest region under different SSP scenarios using three-dimensional simulations"

_EGUsphere, 2025_

## Community Comment (CC2)

**Review Report**

This paper entitle" **Assessing economic impacts of future GLOFs in Nepal's Everest region under different SSP scenarios using three-dimensional simulations**" by Furain and Sauter model the possible future GLOF from existing lakes and possible future glacial lakes in the Nepal's Everest region by using 3D OpenFOAM model and additionally assess the downstream impacts. The paper is suited for NHESS, however, there are several concerns which needs to be addressed and I think it will enhance the quality.

[1] The Everest region is widely studied for glaciers, lakes and GLOF studies. Thus, some of the localized important studies can be considered in the study. For example, glacial lakes in these region among others has undergone highest expansion in the Nepalese Himalaya (Khadka et al., 2018), with numerous GLOF events and notable lakes modeled in this study identified as dangerous (Bajracharya et al., 2020; Khadka et al., 2021). Further, Gouli et al. (2023) has modeled combined GLOF effects of upper and lower Barun lakes in the region. This region has also witnessed number of GLOF events including 2017 from Langmale lake (Byers et al., 2018).

[2] Methods: The methods should be explicitly described as this study used a comparably new model for GLOF simulations. In section 4.2, the authors should briefly illustrate which equations were utilized in the Open foam to simulate the dam break flow and downstream GLOF routing. Lines 220 to 224 – The authors state that the flow in this study involves air and water. It is acknowledged that GLOFs are generally mixture of water and various type of sediments (as mentioned by authors in several places), only considering 2 phase modeling (air and water) will not oversimplify the GLOF rheology? What differences can be found when comparing the results with clear water modeling or others? While this model may offer improvements over traditional clear water flow modeling, it has significant limitations when it comes to accurately simulating the complex processes involved in GLOFs. These events often carry large boulders and debris, as demonstrated by the recent GLOF event from small lake in Thame, Everest, on August 16, 2024. Do GLOF significantly attenuate in those places as shown in Fig 3 or some thresholds were set, please mention? In Fig 3, since lake volumes are large especially for Bhote and Ngojumba Tsho will max inundation depth will be only 20 m, recheck? Author uses a constant manning value, discuss its limitation in discussion.

[3] Impact analysis: Figure 8: There are no roads in Phakding. I feel authors are confused with walking/trekking trails as there are no roads in Everest region. Recently, earthen road was constructed up to Surkhe. This must be corrected throughout the text and impact analysis, otherwise it will exaggerate the economic analysis. Local Impacts: Further, it would add a value to assess impacts to tourism trekking routes and discuss briefly about

the direct-indirect impacts on tourism, local economy and jobs…as these places are touristic hub. Imja on Sagarmatha National Park, Barun Tsho on Makalu Barun National Park, Tsho Rolpa on Gaurishankar Conservation Area. Glacial lakes provide opportunity for adventure trekking and significantly engage locals for economic opportunities (see Khadka et al., 2025) which might be disrupt due to GLOFs leading to huge socio-economic setback.

[4] Discussion: It would also be better to briefly discuss about the uncertainty of formation of potential lakes' dam in Ngozumpa and Bhotekoshi glaciers. Formation of new glacial lakes are not only tied to glacier loss but whether the existing moraine dams the lake or not. Significant drainage of supraglacial lakes (Benn et al., 2001) has breached lateral moraine of Ngozumpa glacier and also future increase in debris cover in the glaciers might inhibit the ongoing melting.

L15 too many uses of between

L25 Also refer (Khadka et al., 2025)

Figure 1: Are the black dots on future lake extent supraglacial lakes? The location of Mt Everest peak is incorrect

L126 More studies have studied GLOF modeling from Lower Barun/Tsho Rolpa

L139 Are BhoteTsho and Ngozumpa glacial lakes susceptible to possible mass movements, do they have enough topographical potential to hit lakes directly? Since modeled lakes are large, the volumes of most glacier avalanche that trigger GLOFs in Himalaya and Tibetan Plateau are about 25–50% of the volume of the lake (Yu et al., 2023).

L168 Since the results are based on depth damage curves, it would be essential to show how it was calculated, authors can use supplementary? Which year rates were used, are they latest?

L185 What is ArcGIS surface reflectance data? Which satellite data you used?

L207 Is there such ArcGIS satellite imagery? Name of images can have referred directly. Further, these maps need to be latest to confirm the latest data. Which date OSM data you used, OSM data are regularly updated

L233 elected or selected? Please use correct word throughout the paper

L235 Which satellite imagery?

L265 Eq. 2 Explain what those coefficient means

5.1 Overview is not results, replace with appropriate title

Figure 3, legend: replace comma with decimal. What does cross section mean? Make the symbol of places hollow or place it bit side so that inundation can be seen.

L515 citations needed. Please confirm whether Imja have EWS or not? I think it does not have

Figure 7/8/9 did authors modify the extent (edges)?

**References:** These references serve as examples from review process without the need to adhere strictly to the authors for citation purposes.

Bajracharya SR, Shrestha AB, Shrestha F, Wagle N, Maharjan SB, Sherpa TC. Inventory of glacial lakes and identification of potentially dangerous glacial lakes in the Koshi, Gandaki, and Karnali river basins of Nepal, the Tibet autonomous region of China, and India. International Centre for Integrated Mountain Development (ICIMOD); United Nations Development Programme (UNDP), Kathmandu, Nepal, 2020, pp. 54.

Benn D, Wiseman S, Hands K. Growth and drainage of supraglacial lakes on debris mantled Ngozumpa Glacier, Khumbu Himal, Nepal. Journal of Glaciology 2001; 47: 626-638 DOI: https://doi.org/10.3189/172756501781831729.

Byers AC, Rounce DR, Shugar DH, Lala JM, Byers EA, Regmi D. A rockfall-induced glacial lake outburst flood, Upper Barun Valley, Nepal. Landslides 2018: 1-17.

Gouli MR, Hu K, Khadka N, Talchabhadel R. Hazard assessment of a pair of glacial lakes in Nepal Himalaya: unfolding combined outbursts of Upper and Lower Barun. Geomatics, Natural Hazards and Risk 2023; 14: 2266219 DOI: https://doi.org/10.1080/19475705.2023.2266219.

Khadka N, Chen X, Yong N, Thakuri S, Zheng G, Zhang G. Evaluation of Glacial Lake Outburst Flood susceptibility using multi-criteria assessment framework in Mahalangur Himalaya. Frontiers in Earth Science 2021; 8: 748 DOI: https://doi.org/10.3389/feart.2020.601288.

Khadka N, Liu W, Shrestha M, Watson CS, Acharya S, Chen X, et al. Multidisciplinary perspectives in understanding Himalayan glacial lakes in a climate challenged world. Information Geography 2025; 1: 100002 DOI: https://doi.org/10.1016/j.infgeo.2025.100002.

Khadka N, Zhang G, Thakuri S. Glacial lakes in the Nepal Himalaya: Inventory and decadal dynamics (1977–2017). Remote Sensing 2018; 10: 1913 DOI: https://doi.org/10.3390/rs10121913.

Yu B, He Y, Ye P. Quantitative susceptibility assessment of the breach of moraine-dammed lakes due to glacier avalanches. Cold Regions Science and Technology 2023; 206: 103749 DOI: https://doi.org/10.1016/j.coldregions.2022.103749.

---

## Author Response (AR1)

**Revision for manuscript NHESS 2025-50**

**"Assessing economic impacts of future GLOFs in Nepal's Everest region under different SSP scenarios using three-dimensional simulations"**

**Authors:** Wilhelm Furian, Tobias Sauter

We would like to thank both referees as well as Mr. Khadka for their insightful and helpful comments. We believe it was possible to substantially improve our manuscript based on their input.

In the following sections, we list all of the reviewers' comments (blue). Our responses (black) are also listed with line numbers, which correspond to the track-changes manuscript. Hopefully, this will make it easier to track the revisions. Where necessary, we provide tab-indented citations from the new manuscript to clarify the changes.

**Referee 1:**

Strikingly, the breach scenarios (Table 1) are defined regardless moraine dam geometry, physical limits of breach development, internal structure and possibly overdeepened bedrock terrain. Why 30, 60 and 90 m? Why not 20, 40 or 60 m? Or 10, 20 and 30 m? It is important to highlight that anyhow sophisticated modelling outcomes are totally dependent on rather arbitrary definition of these breach scenarios. It is also important to highlight that these scenarios have different probabilities for individual studied lakes and that some are not even realistic.

You are absolutely right in pointing out that our rationale for choosing these exact breach parameters was not sufficiently explained. We have rewritten the respective paragraphs (L202-222) to include the following explanations:

Due to the requirements of working with OpenFOAM, several concessions have to be made when creating breach scenarios. As we are working with this 3D numerical model, several options that can be included in 2D approaches are not available to us. We cannot define a desired hydrograph for each lake or couple a designated breach model to the simulation. Rather, the hydrograph in OpenFOAM results from the breach scenario and the lake volume and is not specified directly. Therefore, in our approach, the DEM for each simulation run, including the moraine breach, has to be manually created and transformed into an STL surface. As we ran almost 100 simulations, it would not be feasible to define individual moraine breaches for every lake in every scenario, taking into account, e.g., the changing moraine structure, the growing lake volume and depth etc.

We fully agree that the magnitude of the simulated floods mostly depends on the chosen breach scenarios and mention this in Table 4. However, we recognize the need to further explain the reasoning behind our choices and argue for their validity:

Since we chose to use a 3D model to improve the simulation accuracy, we had to limit the number of breach scenarios in order not to increase the computational time beyond the 10-12 weeks that the simulations already had to run (excluding the time needed to set-up the simulations). The chosen breach scenarios follow published replication studies of historic GLOFs and other GLOF simulation studies. We excluded the lower and higher estimates of some of those studies, as the higher ones seem unrealistic and the impact of the lower scenarios would be too small to justify the longer computation time. This is how we chose our two main scenarios: 30 and 60 m, and a third extreme depth of 90 m.

The 30m breach was chosen because it represents a significant, but not extreme, moraine breach. The 60m breach was chosen as the upper limit of GLOF events (as seen at South Lhonak Lake, 10.1126/science.ads2659). The 90m scenario was chosen as a potential GLOF of extreme magnitude, theoretically possible at all lakes except Imja Tsho. However, due to its unlikely nature, we did not evaluate this scenario further. Instead, we provide the results in the supplementary material.

As a degree of generalization is required for large-scale OpenFOAM simulations, we use these breach parameters for all five lakes. Apart from the fact that it would not be feasible to individually create different moraine breaches for our 3D approach, many of the parameters needed to delineate individual breaches are not easily quantifiable for the future scenarios: Internal moraine ice can melt, and moraines can be damaged by earthquakes or lowered by internal piping, etc. We do not aim to investigate the probabilities of specific events, as it would be beyond the scope of our study to approximate the necessary parameters for the whole of the 21st century. However, this is not made sufficiently clear in our manuscript and we thank you for pointing this out. Our manuscript now includes a clearer description and a more substantial justification for our choice of breach parameters.

Now what is called BR1 (lower boundary; 30 m breach depth) is already pretty harsh scenario and the term "lower boundary" is misleading in this context. How many examples of 30 m deep breaches do we have from lakes of similar size and topographic setting? I don't think about many. The BR2 (upper boundary; 60 m breach depth) is not only unlikely but also unrealistic for lakes with flat and wide dam geometry (such as Imja which dam height is 55 m, according to 10.5194/hess-29-733-2025 or 35 m according to 10.5194/hess-19-1401-2015).

We agree that the phrase "lower boundary" is misleading for a 30m breach, as it may well be the lower boundary in this study, but not in the wider scientific context. We have rephrased the relevant paragraph (L211-222) as follows:

> To maintain computational feasibility, we did not simulate smaller breach scenarios; the BR1 scenario already represents a moraine incision of 30 meters. This is consistent with several reconstruction studies that model historic GLOFs with moraine breaches between ~25 and ~35 meters (Watanbe and Rothacher, 1996; Somos-Valenzuela and McKinney, 2011; Nie et al., 2020; Mergili et al., 2020; Zheng et al., 2021). While such a breach would constitute a significant moraine failure, larger events have been documented. Accordingly, BR1 is referred to as a "medium" scenario in this study. The second scenario, BR2, categorized as "high" in this study, reflects the upper range of observed breach dimensions, such as those reported at South Lhonak Lake during the Sikkim flood of 2023 (Sattar et al., 2025). Finally, the third scenario (BR3) represents a theoretical breach of "extreme" magnitude. However, following initial trial runs, we excluded BR3 from the main analysis due to the unrealistically large flood volumes it produced. Although similar breach dimensions have been used in other studies (e.g., Sattar et al., 2021; Mandal et al., 2025), we judged such scenarios to be extraordinarily large. Consequently, the main analysis is based on BR1 and BR2, except at Imja Tsho, where only BR1 was applied due to the moraine's lower maximum height. BR3 results are presented in Tables S1–S3 to illustrate potential worst-case outcomes of GLOFs of extreme magnitude.

You are absolutely right, a 60m breach would be very unlikely at Imja Tsho, as it would represent more than a complete moraine incision. We used an approach similar to the recently published 10.5194/hess-29-733-2025, where "the maximum breach depth is considered to reach the marine dam's maximum height and extend from the dam crest down

to the point where the hummocky terrain ends" (p. 737). However, we have failed to explain this in detail and will therefore adapt our manuscript to exclude the 60m scenario at Imja.

In Chapter 6.1, we now discuss the performance of our model in comparison to previous GLOF simulations with moraine breaches of the same magnitude (L513-533):

> CFD modeling is not without its own set of uncertainties, mainly regarding the resolution of the mesh, which is, in this case, somewhat coarser than the resolution of the DEM. To place our results in a larger scientific context, we compared our results with previous GLOF simulations at Tsho Rolpa, Lower Barun Lake and Imja Tsho. Despite the different modeling approaches, our results generally align with these studies. A visual comparison with the results of Sattar et al. (2021) reveals similar flood extent and inundation patterns at Lower Barun Lake, although our simulations produce slightly higher flow velocities and inundation depths at certain downstream locations. Mandal et al. (2025) report comparable maximum inundation depths of approximately 20 meters for a 50-meter breach, which aligns well with the results from our 60-meter BR2 scenario. Reported flow velocities ranging from 3 to 10 m/s are also in good agreement with our estimates, which fall between 3 and 8 m/s.
>
> At Tsho Rolpa, Chen et al. (2022) simulated a GLOF from a 30-meter moraine breach reaching as far as Manthali—closely matching the run-out distance of our BR1 scenario. However, the higher discharges in our model lead to higher velocities. Several other studies report discharge and inundation depths consistent with our findings: Shrestha et al. (2012) estimated a peak discharge of 90,000 $m^3\ s^{-1}$ in their highest scenario, compared to the 81,000 $m^3\ s^{-1}$ in our simulation. Chen et al. (2025) reported peak discharges between 13,000 and 15,000 $m^3\ s^{-1}$, which aligns well with our value of approximately 12,300 $m^3\ s^{-1}$. Both discharge and inundation depth estimates are in good agreement with the study by Kayastha and Maskey (2024), which simulated GLOFs from moraine breaches with a width of 20 meters and 40 meters. At Imja Tsho, Somos-Valenzuela et al. (2015) compared different lake lowering scenarios, which could account for their lower discharge and GLOF run-out estimations. Chen et al. (2025) estimated a mean discharge at Imja Tsho of 15,000 $m^3\ s^{-1}$, which aligns well with the 11,800 $m^3\ s^{-1}$ in our model.

For a comparison, the breach which developed during the 2023 South Lhonak GLOF – the largest GLOF from a moraine-dammed lake in High Mountain Asia in past decades – is 55 m deep (see 10.1126/science.ads2659) . The use of as extreme scenario as BR3 (90 m breach depth) needs special justification on a case-by-case basis.

We have already mentioned the unrealistic nature of the 90m breach in the manuscript, but we have rephrased L217-L222 to emphasize that we are excluding it from our analysis for this reason.

**Referee 2:**

We appreciate your detailed suggestions and have implemented your advice regarding additional references, improvements in wording and style, as well as clarifications where necessary. Below, we address your specific comments in detail:

The Everest region is widely studied for glaciers, lakes and GLOF studies. Thus, some of the localized important studies can be considered in the study. For example, glacial lakes in these region among others has undergone highest expansion in the Nepalese Himalaya (Khadka et al., 2018), with numerous GLOF events and notable lakes modeled in this study identified as dangerous (Bajracharya et al., 2020; Khadka et al., 2021). Further, Gouli et al. (2023) has modeled combined GLOF effects of upper and lower Barun lakes in the region. This region has also witnessed number of GLOF events including 2017 from Langmale lake (Byers et al., 2018).

We have incorporated more studies into the first paragraph of Chapter 2 to improve the representation of previous scientific efforts (L100-105).

[2] Methods: The methods should be explicitly described as this study used a comparably new model for GLOF simulations. In section 4.2, the authors should briefly illustrate which equations were utilized in the Open foam to simulate the dam break flow and downstream GLOF routing.

We chose not to include additional equations from OpenFOAM in the manuscript, as they are both highly numerous and not directly specific to our study. However, to support interested readers, we added a reference to the OpenFOAM user manual, where the relevant equations and their interrelations are comprehensively described. The dam break flow modeling approach is detailed in Chapter 4.2 and in Table 1 of the manuscript. There, we describe the relation between different breach scenarios, their corresponding opening times, and the use of the baffle method to simulate progressive breach formation. Following the suggestions of other reviewers, we have rewritten some of these paragraphs to better explain our choices.

Lines 220 to 224 – The authors state that the flow in this study involves air and water. It is acknowledged that GLOFs are generally mixture of water and various type of sediments (as mentioned by authors in several places), only considering 2 phase modeling (air and water) will not oversimplify the GLOF rheology? What differences can be found when comparing the results with clear water modeling or others?

We agree with your observation that simulating GLOFs as hyperconcentrated flows should lead to improved realism compared to traditional clear-water models. We also recognize the limitations inherent to our approach and have discussed some of these in Chapter 6.1. Following the suggestions of previous reviewers as well as your own, we have expanded this discussion further to better highlight the limitations and uncertainties (L289-303).

While this model may offer improvements over traditional clear water flow modeling, it has significant limitations when it comes to accurately simulating the complex processes involved in GLOFs. These events often carry large boulders and debris, as demonstrated by the recent GLOF event from small lake in Thame, Everest, on August 16, 2024.

Regarding your concern: The computational demands of a three-dimensional two-phase GLOF simulations are already considerable. While multiphase solvers exist within

OpenFOAM, they are primarily designed for small-scale industrial processes, such as multiphase flows in oil extraction or nuclear reactors. Using a three-phase solver to model GLOFs at a large spatial scale would not have been feasible with the computational resources available to us. Tracking three interfaces (air, water, sediment/boulders) would significantly increase computational complexity, particularly in terms of interface tracking within the VOF method. We explain this in L296-301:

> While incorporating an explicit sediment transport or a multiphase model could improve process detail even further, it would significantly increase computational demands—especially given the already long runtimes of high-resolution two-phase simulations. Even with HPC resources, it is not feasible to resolve GLOF processes ranging from the transport of fine sediment to the movement of large boulders within a single simulation framework of this scale. We therefore model outburst floods within a two-phase framework, which allows us to capture the essential dynamics while maintaining computational feasibility.

In Fig 3, since lake volumes are large especially for Bhote and Ngojumba Tsho will max inundation depth will be only 20 m, recheck?

You are right in that larger lakes have the potential to produce GLOFs capable of causing massive inundation depths. However, we do not use a physical model to simulate the moraine incision, but rather employ a parametric scenario-based approach and provide the breach parameters to OpenFOAM to simulate the according hydrograph.

Therefore, in our study, the lake volume represents the potential for a GLOF's magnitude, while the breach size is responsible for realizing this potential. Since our breaches are of the same magnitude for all lakes, the inundation depths of all lakes are of the same order of magnitude. Please also see our response to Adam Emmer's valuable comment above, where we supply previously lacking information on our reasoning behind the breach parameter selection.

Author uses a constant manning value, discuss its limitation in discussion.

Regarding surface roughness, we employed a constant Manning value across the domain. Given the scarcity of high-resolution surface roughness data—especially for projecting into future conditions—this approach follows common practice in GLOF modeling studies. While Manning's n may vary between mountainous and more lowland river reaches, implementing spatially variable roughness would require splitting the computational mesh into multiple patches or implementing customized boundary conditions, both of which would significantly increase computational time. We explain this in L 308-314:

> We employ a constant surface roughness value across the computational domain given the scarcity of high-resolution surface roughness data—especially for projecting into future conditions. With this approach, we follow numerous previous GLOF simulations studies (e.g., Larocque et al., 2013; Westoby et al., 2015; Azeez et al., 2020; Majeed et al., 2021; Yang et al., 2023). While the terrain roughness may vary between mountainous and more lowland river reaches, implementing spatially variable roughness values would require splitting the computational mesh into multiple patches or implementing customized boundary conditions, both of which would significantly increase computational time.

Impact analysis: Figure 8: There are no roads in Phakding. I feel authors are confused with walking/trekking trails as there are no roads in Everest region. Recently, earthen road was constructed up to Surkhe. This must be corrected throughout the text and impact analysis, otherwise it will exaggerate the economic analysis.

You are absolutely right, the classification of every trail and trek as "roads" is misleading and was changed in the relevant figures and paragraphs.

Regarding the impact on the socio-economic assessment: The lack of detailed classification in the OSM-roads data prevented an in-depth assessment of the financial impact of the destruction of infrastructure. Therefore, we unfortunately had to refrain from providing monetary values associated with the inundated routes. Instead, we point out the aggravating nature of road/trek destruction during a GLOF event as it hinders post-disaster assistance.

Local Impacts: Further, it would add a value to assess impacts to tourism trekking routes and discuss briefly about the direct-indirect impacts on tourism, local economy and jobs…as these places are touristic hub. Imja on Sagarmatha National Park, Barun Tsho on Makalu Barun National Park, Tsho Rolpa on Gaurishankar Conservation Area. Glacial lakes provide opportunity for adventure trekking and significantly engage locals for economic opportunities (see Khadka et al., 2025) which might be disrupt due to GLOFs leading to huge socio-economic setback.

We appreciate your comments regarding the potential impact of GLOFs on the tourism sector. While a detailed examination is beyond the scope of our work, in L623-630, we now discuss this aspect:

> Aside from these immediate impacts, all simulated GLOFs have the potential to disrupt local tourism by damaging critical infrastructure such as bridges and hiking trails. Disruption of these structures would not only hinder tourist flows but also inflict ripple effects across local economies reliant on trekking services, accommodation, and guiding (Nyaupane and Chhetri, 2009). This vulnerability is a growing concern, especially in the Everest Region, where a positive correlation has been found between employment in the tourism industry and increased awareness of GLOF risk (Sherpa et al., 2019). These results underscore the need for multidisciplinary local adaptation strategies that integrate both vulnerabilities and economic opportunities linked to tourism within comprehensive GLOF-risk management frameworks (Khadka et al., 2025).

Discussion: It would also be better to briefly discuss about the uncertainty of formation of potential lakes' dam in Ngozumpa and Bhotekoshi glaciers. Formation of new glacial lakes are not only tied to glacier loss but whether the existing moraine dams the lake or not. Significant drainage of supraglacial lakes (Benn et al., 2001) has breached lateral moraine of Ngozumpa glacier and also future increase in debris cover in the glaciers might inhibit the ongoing melting.

We agree that the formation of glacial lakes depends on a wide range of factors that cannot be predicted with certainty. We acknowledge this complexity in the revised manuscript (L130-133), while a comprehensive discussion of all relevant processes is beyond the scope of our study, as these topics are addressed more fully in previous specialized research (Furian et al., 2021; Furian et al., 2022).

Figure 1: Are the black dots on future lake extent supraglacial lakes? The location of Mt Everest peak is incorrect

The dark blue areas on the glaciers indicate existing supraglacial lakes, and appear black due to their outline. We have clarified this in the caption. The label "Mt Everest" was intended to show the general location of the mountain, rather than the exact position of the summit. However, we agree that its placement could be improved and adjusted it slightly.

L126 More studies have studied GLOF modeling from Lower Barun/Tsho Rolpa

We have added more studies to better represent previous scientific efforts.

L139 Are BhoteTsho and Ngozumpa glacial lakes susceptible to possible mass movements, do they have enough topographical potential to hit lakes directly? Since modeled lakes are large, the volumes of most glacier avalanche that trigger GLOFs in Himalaya and Tibetan Plateau are about 25–50% of the volume of the lake (Yu et al., 2023).

For the topographical potential, we refer to our previous publications (Furian et al., 2021; Furian et al., 2022), where the surrounding slopes of potential glacial lakes and the lakes' evolution are discussed in detail. In this context, Ngojumba Tsho shows a slightly higher predisposition for mass movements than Bhote Tsho. However, as our study does not model specific GLOF triggers, we cannot provide estimates on potential avalanche volumes. We agree that this would be an interesting subject for future research.

L168 Since the results are based on depth damage curves, it would be essential to show how it was calculated, authors can use supplementary? Which year rates were used, are they latest?

The depth-damage relationship is described at the end of Chapter 3 and the end of Chapter 4.1, where we explain the different damage classes and provide references for further reading. The curves are provided by the Joint Research Centre (JRC) of the European Union in 2017 (Huizinga et al., 2017). To maintain the focus of the manuscript, we prefer not to reproduce the full damage-depth curves, as they are freely available in the original publication.

L185 What is ArcGIS surface reflectance data? Which satellite data you used?

We now mention the high-resolution Maxar data, which was provided as a basemap by ArcGIS.

L207 Is there such ArcGIS satellite imagery? Name of images can have referred directly. Further, these maps need to be latest to confirm the latest data. Which date OSM data you used, OSM data are regularly updated.

We have added the clarification that Maxar data was used. And you are right, we were missing the correct citation of the OSM data, which has been added.

L235 Which satellite imagery?

We have added the clarification that Maxar data was used

L265 Eq. 2 Explain what those coefficient means

The empirical coefficients in Equation 2 originate from O'Brien and Julien (1988) and were derived through regression analysis by those authors. They define the relationship between yield stress, viscosity, and the volumetric sediment concentration. For a detailed discussion, we refer readers to the original publication.

5.1 Overview is not results, replace with appropriate title

Chapter 5.1 is titled "Overview" because it summarizes the results of the GLOF simulations. Given the large number of model runs, we believe an introductory chapter providing a concise overview before discussing individual aspects in detail is necessary. We therefore consider the title appropriate.

Figure 3, legend: replace comma with decimal. What does cross section mean? Make the symbol of places hollow or place it bit side so that inundation can be seen.

Thank you for the comments on Figure 3. The legend was corrected and the cross-section names were moved so that the inundation can be seen clearer.
As described in the first paragraph of Chapter 5.1, the term "cross-section" refers to a vertical slice through the computational mesh along the GLOF path, perpendicular to the primary flow direction. At these cross-sections, we assess key parameters such as valley discharge, inundation depth, etc.

L515 citations needed. Please confirm whether Imja have EWS or not? I think it does not have

It is hard to discern whether the EWS at Imja Tsho is still functional. This 2021 EGU presentation (10.5194/egusphere-egu21-4163) and this 2022 study (10.1016/j.ijdrr.2022.102914) make no mention of it being discontinued. Therefore, if you could provide additional information on the status of the EWS at Imja Tsho, we would be very grateful.

Figure 7/8/9 did authors modify the extent (edges)?

Unfortunately, it is not completely clear what this question is about. In general, we would like to reinforce that the figures of course present the unmodified simulation results after postprocessing and integration into ArcGIS. No manual adjustments to the flood extent were made. The edges of the figures show the grid, which is different for each image due to the orientation (see north arrows).

References

[revised manuscript text omitted]